# Genomic Insights into Colistin and Tigecycline Resistance in ESBL-Producing *Escherichia coli* and *Klebsiella pneumoniae* Harboring *bla*_KPC_ Genes in Ecuador

**DOI:** 10.3390/antibiotics14020206

**Published:** 2025-02-17

**Authors:** David Ortega-Paredes, Felipe Del Canto, Rafael Rios, Lorena Diaz, Jinnethe Reyes, Cesar A. Arias, Jeannete Zurita

**Affiliations:** 1Facultad de Ciencias Médicas Enrique Ortega Moreira, Carrera de Medicina, Universidad Espíritu Santo, Samborondón 092301, Ecuador; daortegap01@gmail.com; 2Programa de Microbiología y Micología, Instituto de Ciencias Biomédicas, Facultad de Medicina, Universidad de Chile, Santiago 9170022, Chile; 3Molecular Genetics and Antimicrobial Resistance Unit, International Center for Bacterial Genomics, Universidad El Bosque, Bogotá 111321, Colombia; 4Division of Infectious Diseases and Center for Antimicrobial Resistance and Microbial Genomics, McGovern Medical School, University of Texas Health Science Center, Houston, TX 77030, USA; 5Center for Infectious Diseases, School of Public Health, University of Texas Health Science Center, Houston, TX 77030, USA; 6Unidad de Investigaciones en Biomedicina, Zurita & Zurita Laboratorios, Quito 170104, Ecuador

**Keywords:** *Escherichia coli*, *Klebsiella pneumoniae*, colistin resistance, tigecycline resistance, *bla*
_KPC_, *mcr*-1, Ecuador

## Abstract

**Introduction: ***Escherichia coli* (*E. coli*) and *Klebsiella pneumoniae* (*K. pneumoniae*) are resistant to third-generation cephalosporins (3GCs), carbapenems, colistin, and tigecycline, making them a major public health priority, mainly within the developing world. However, their genomic epidemiology and possible determinants of resistance remain to be elucidated. Thus, this study aimed to perform a genomic characterization of *E. coli* and *K. pneumoniae*, both of which are resistant to last-line antibiotics, isolated from humans, poultry, and a dairy farm environment within Ecuador. **Methods:** This study analyzed nine 3GC-resistant *E. coli* isolates harboring the *mcr*-1 gene (six from poultry farms, two from human infections, and one from dairy farm compost), together with ten isolated colistin- and carbapenem-resistant *K. pneumoniae* clinical samples. **Results:** The *E. coli* isolates of human origin belonged to ST609 and phylogroup A, while the poultry and compost isolates belonged to phylogroups A, B1, E, and F. Diverse STs of the *K. pneumoniae* isolates included ST13 (five isolates), ST258 (four isolates), and ST86 (one isolate). Within the *E. coli* isolates, *bla*_CTX-M-55_, *bla*_CTX-M-65_, *bla*_CTX-M-15_, and *bla*_CTX-M-2_ genes were identified. This study also identified *bla*_CMY-2_ and *bla*_KPC-3_ (the latter in a carbapenem-susceptible isolate). In *E. coli*, the plasmid-borne *mcr*-1.1 gene was identified across all *E. coli* isolates within an IncI2 plasmid. Tigecycline-reduced susceptibility or resistance was related to missense amino acid substitutions coded in the *marA* and *acr*A genes. Within *K. pneumoiae*, *bla*_CTX-M-15_ and *bla*_CTX-M-65_, on the one hand, and *bla*_KPC-2_ and *bla*_KPC-3_, on the other, were associated with 3GC and carbapenem resistance, respectively. The *bla*_KPC-2_ allele was identified in a ~10 kb Tn*4401* transposon (*tnpR–tnpA–istA–istB–bla_KPC-2_–tnpA*). In *K pneumoniae*, sequence data and phenotypic analysis linked a nonsense amino acid substitution coded in the *mgrB* (K3*) gene and missense amino acid substitutions coded in the *marA*, *acr*A, *arnB*, *eptA*, *pmrB*, *pmrJ*, and *phoQ* genes to colistin resistance. Meanwhile, tigecycline resistance was linked to nonsense and missense amino acid substitutions coded within the *ramR* sequence. Additionally, this study identified several integron structures, including Int191 (*5′CS-dfrA14-3′CS*), which was the most prevalent integron (Int) among *E. coli* and *K. pneumoniae* isolates in this study, followed by Int0 (*5′CS-3′CS*) and Int18 (*5′CS-dfrA1-3′CS*). **Conclusions:** These results contribute to the genomic epidemiology of MDR *E. coli* and *K. pneumoniae* in our setting and to the worldwide epidemiology in the One Health approach.

## 1. Introduction

In recent years, the emergence of *Enterobacterales* strains that are resistant to all β-lactam antibiotics has restricted therapeutic options to only last-line antibiotics, including colistin and tigecycline [1,2]. Furthermore, the emergence of *Escherichia coli* (*E. coli*) and *Klebsiella pneumoniae* (*K. pneumoniae*) strains that are resistant to multiple antibiotics, including third-generation cephalosporins (3GCs), carbapenems, colistin, and tigecycline, has markedly affected the ability to treat patients, particularly within developing countries [3].

The introduction of 3GCs as a response to the increasing resistance to classic antibiotics (first generations of β-lactams, tetracyclines, quinolones, folate pathway inhibitors, and sulfonamides) has provided a degree of hope with respect to the treatment of complex infections driven by *Enterobacterales*. However, the extended use of 3GCs to treat human and animal infections has led to their utility becoming compromised, mostly due to the emergence of extended-spectrum β-lactamase (ESBL) genes [4,5]. Consequently, in response to the dissemination of the ESBL genes *bla*_TEM_, *bla*_SHV_, and *bla*_CTX-M_ among *Enterobacterales*, carbapenems have become the gold standard for treating multidrug-resistant (MDR) infections. Unfortunately, during the last decade, the acquisition of genes coding for enzymes capable of hydrolyzing carbapenems (e.g., *bla*_KPC_ and *bla*_NDM_) has become an urgent public health issue, jeopardizing the treatment of severe infections caused by *Enterobacterales* [5].

The lack of reliable therapeutic options to treat carbapenem-resistant *Enterobacterales* (CRE) has led to the utilization of older compounds, such as polymyxins. Indeed, colistin (polymyxin E) has been employed in veterinary medicine for decades, mainly for prophylaxis and the treatment of infections caused by Enterobacterales and also as a growth-promoting agent in some countries [6,7]. Since polymyxins have been employed against CRE, resistance has gradually emerged (mostly due to amino acid substitutions coded within chromosomal genes) [8], becoming a major public health priority, particularly within developing countries [9]. Furthermore, reports on mobile elements carrying colistin resistance determinants (e.g., mobile colistin resistance, *mcr* genes) are scarce [10].

In 2005, tigecycline was approved by the Food and Drug Administration (FDA) for the treatment of skin/soft tissue and intra-abdominal infections [11]. Due to tigecycline’s in vitro activity against CRE, this antibiotic is typically employed as salvage therapy for CRE [12] (mostly in combination with other agents), with resistance in *Enterobacterales* remaining minimal. Nonetheless, resistance to tigecycline has been characterized as involving mutations in ribosomal proteins and the presence of *tet*(A) and *tet*(M) efflux pumps, among others [13]. Recently, the presence of the mobile determinant *tet*X4, coding for a tetracycline-inactivating enzyme, has been described as driving tigecycline resistance within animal isolates [14]. In Ecuador alone, *bla*_CTX-M_, *bla*_KPC_, and *mcr*-1 genes in *E. coli* have been reported within isolates from humans, food, poultry, pigs, dog feces, and the environment [15,16,17,18,19,20,21,22,23,24,25]. However, this issue needs research and community communication strategies [26]. Furthermore, carbapenem-resistant *K. pneumoniae* was reported in Ecuador in 2010 [27], with colistin-resistant *K. pneumoniae* required to be reported through the Ecuadorian national surveillance program and reported in accordance with the One Health approach [28]. However, the genomic epidemiology of *Enterobacterales* resistant to colistin, a last-resort antibiotic among other new alternatives, together with potential determinants of resistance, remains to be elucidated.

Consequently, the aim of this study was to perform genomic characterization of *E. coli* and *K. pneumoniae* within human, poultry, and environmental isolates that were resistant to last-line antibiotics within Quito, Ecuador.

## 2. Results

### 2.1. Antibiotic Susceptibility Profile

The antibiotic susceptibility profiles are summarized in Table 1. The isolates (nine *E. coli* and ten *K. pneumoniae*) exhibited resistance to colistin, with MIC values ranging from 4 to ≥16. All *E. coli* isolates were resistant to ampicillin and ceftriaxone, and all, excluding one isolate, were resistant to ciprofloxacin. Two *E. coli* isolates (1409 and 4536) exhibited reduced susceptibility to tigecycline, and all were susceptible to carbapenems and amikacin. The majority of the *K. pneumoniae* isolates (excluding one isolate—238) exhibited resistance to most antibiotics that were analyzed, with reduced susceptibility to tigecycline (Table 1 and Figure 1).

### 2.2. Acquired Resistance Genetic Determinants and Mutations Related to Antibiotic Resistance Profiles

Among the ESBL genes, *bla*_CTX-M-55_, *bla*_CTX-M-65_, *bla*_CTX-M-15_, and *bla*_CTX-M-2_ were identified. Additionally, *bla*_CMY-2_ and *bla*_KPC-3_ (the latter within a carbapenem-susceptible isolate) were annotated. The plasmid-borne *mcr*-1.1 gene was identified across all *E. coli* isolates. The isolates with resistance or reduced susceptibility to tigecycline harbored missense mutations within the efflux system regulatory genes *marA* and *acrA*.

The *K. pneumoniae* isolates harbored the ESBL genes *bla*_CTX-M-15_, *bla*_CTX-M-65_, *bla*_KPC-2_, and *bla*_KPC-3_, encoding serin-carbapenemases. The data analysis suggested a nonsense mutation in *mgr*B (K3*), along with several missense mutations in *arnB*, *eptA*, *pmrB*, *pmrJ*, and *phoQ,* which are linked to colistin resistance and are involved in the lipid A-Ara4N pathway. Tigecycline resistance in the *K. pneumoniae* isolates was related to nonsense and missense mutations in the *ramR* gene, the regulatory gene for the system. Additionally, this study identified several integrons, including Int191 (*5’CS-dfrA14-3’CS*), which was the most prevalent integron among the *E. coli* and *K. pneumoniae* isolates, followed by Int0 (*5’CS-3’CS*) and Int18 (*5’CS-dfrA1-3’CS*) (Appendix A).

### 2.3. Phylogenetic Analysis

The *E. coli* isolates from humans belonged to sequence type (ST) 609 and phylogroup A, while the poultry and compost isolates belonged to differing STs and phylogroups A, B1, E, and F. Conversely, the *K. pneumoniae* isolates were ST13 (n = 5), ST258 (n = 4), and ST86 (n = 1). The enterobase-based SNPtree of the *E. coli* isolates revealed that the human samples were closely related and had no genetic relationship with the isolates from compost or poultry. Regarding *K. pneumoniae*, the phylogenetic tree revealed three clusters that were in agreement with the ST (Figure 1 and Appendix A).

### 2.4. Serotypes and Virulence of the E. coli Isolates

The clinical isolates of *E. coli* were serotyped as O9:H4, with the compost isolate identified as H25, while the poultry isolates belonged to the O115:H28, O2:H4, H28, H34, H20, and O157:H21 serotypes. The virulome of the *E. coli* isolates included 14 genes, with each isolate harboring 2–10 genes (Table 2). We clarified the most prevalent virulence gene and the isolate harboring the highest number of virulence genes. The most prevalent virulence gene among the *E. coli* isolates was *gad* (glutamate decarboxylase). Additionally, the isolate with the highest number of virulence genes was isolate 136A, which originated from avian sources and contained the greatest diversity of these genes. However, such virulence factor profiles did not identify human-related intestinal or extraintestinal *E. coli* pathotypes.

### 2.5. Plasmid Analysis of the mcr-1 Gene

The PlasmidFinder^®^ tool was able to identify 1–8 co-existing plasmids within each isolate. Interestingly, plasmids belonging to the IncI2 type were detected across all *E. coli* isolates. Plasmid and locus alignment located the *mcr*-1.1 gene in the IncI2 plasmid, sharing >90% similarity with the 64,015 bp pHNSHP45 plasmid (GenBank nucleotide KP347127.1) (Figure 2A).

All of the *E. coli* J53 transconjugants selected for colistin resistance produced amplicons with *mcr-*1-specific primers [20], and none of them produced amplicons following PCR-based replicon typing (PBRT). Additionally, the *mcr*-1.1 gene was harbored within identical genetic environments for all isolates, where Tnp (*ISApl1*) is absent (Figure 2B).

### 2.6. Genetic Environment of Carbapenem Resistance Genes

The *bla*_KPC-2_ and *bla*_KPC-3_ genes were identified within the transposon Tn*4401* (Figure 3). Genomic segments containing *bla*_KPC-3_ in the five *K. pneumoniae* ST13 isolates exhibited high sequence identity with plasmid pKPC-CAV1042-89 (GenBank nucleotide CP018669), harbored by a *K. pneumoniae* ST244 strain obtained from a patient with a urinary tract infection in Ecuador (Figure 3A, left rings, and Figure 3B). Conversely, the genetic context of Tn*4401* within the four *K. pneumoniae* ST258 isolates harboring *bla*_KPC-2_ showed high homology with the pKPC-484 plasmid (GenBank nucleotide CP008798.1) from another previously described clinical ST258 strain (Figure 3A, right rings, and Figure 3C). Although *E. coli* 136A was susceptible to carbapenems, sequence analysis revealed the presence of the *bla*_KPC3_ gene within its genome.

## 3. Discussion

In this study, all *E. coli* isolates retained susceptibility to amikacin and carbapenems. Conversely, all carbapenem-resistant *K. pneumoniae* isolates were also resistant to amikacin. The human and poultry isolates belonged to phylogroups A and B1. Previously, multi-resistant *E. coli* belonging to these phylogenetic groups were identified within human-dwelling commensal flora [29]. Moreover, such phylogroups are the most frequently isolated within healthy poultry [30].

Interestingly, none of the *E. coli* isolates were classified in phylogroups B2 or D, commonly associated with extraintestinal infections. However, phylogroup F, a less virulent sister group of phylogroup B2—previously found within 14 environmental and clinical samples—was identified in poultry and compost [31]. These datasets suggest that the isolates are not directly related to human infection. This inference was supported by MLST analysis since it did not demonstrate epidemic clones among the *E. coli* isolates. This was not the case with the *K. pneumoniae* epidemic clones ST13 (n = 5) and ST258 (n = 3), which were highly represented. *K. pneumoniae* ST13 and especially ST258 have been reported globally as important XDR pathogens for humans and as being a major source of nosocomial infections [32,33,34]. Several serotypes harboring between 2 and 10 virulence factors were identified within *E. coli* isolates. However, none of the isolates presented serotypes typically involved in infections or a classic combination of virulence factors linked to intestinal or extraintestinal pathogenic status. Nonetheless, within the poultry isolates, the *hly*F, *iss*, and *iro*N genes, coding for siderophores, were detected. These genes have been previously reported as highly prevalent in ESBL-producing *E. coli* isolated from poultry, and it is suspected that they have a role in human extraintestinal infections.

Moreover, the *iut*A and *omp*T genes—frequently found in *E. coli*—were not detected in this study. Colistin (polymyxin E) is an antibiotic of critical importance to human health. Consequently, its use has recently been restricted in most countries due to complex bacterial infections in humans [35]. The use of colistin as a feed additive for animals was banned in China in 2016, following the first report of the *mcr*-1 gene being correlated with food animal production [36]. This initiative was followed by other nations. Within Ecuador, the use of colistin on farms was banned in 2019 [37]. However, reports of colistin-resistant *Enterobacterales*, especially *E. coli*, have increased in recent years [38]. Such reports mostly associate colistin resistance with plasmids carrying *mcr* genes via horizontal gene transfer [39].

All plasmids reconstructed in this study were highly homologous, suggesting that IncI2-type plasmids are important platforms for the spread of *mcr*-1.1 in Ecuador. Previously, IncI2-type plasmids were identified as the most frequent disseminators of *mcr*-1 globally [40,41]. Within these plasmids, *mcr*-1 is typically associated with insertion sequences that allow its transposition to other plasmids or the chromosome.

This genetic feature was previously reported in Japan (GenBank accession, AP019686.1). Similar to previous reports, colistin resistance in *K. pneumoniae* isolates (MIC ≥ 16) from Ecuador was not linked to *mcr* genes. However, this study identified chromosomal mutations that are likely to be involved in lipid A modifications. The first pathway of this resistance includes a mutation in the *mgrB* gene (3K*) that upregulates the two-component regulatory systems, namely, *PhoP/PhoQ and pmrA/pmrB*. The *phoP/phoQ* system indirectly activates *pmrA/pmrB*, conferring colistin resistance by regulating *pmrCAB* expression (transfer of phosphoethanolamine transferase, PEtN-to-lipid A) and *pmrHFIJKLM* (transfer of L-Ara4N to lipid A) operons [42]. The second pathway involves missense mutations within the *arnA*, *phoQ*, *arnT*, *eptA*, *pmrB, and pmrJ* genes that are also part of the regulon for the two-component systems mentioned above. Currently, it is known that a complex combination of mutations in TCRS leads to elevated MIC levels for colistin resistance [43].

This study also found that *E. coli* human isolates exhibit reduced susceptibility to tigecycline. Within such isolates, mutations in *marA* (S103G, H137Y) and *acrR* (V29fs) were identified. Similarly to tetracyclines, tigecycline binds to 30S subunits and interrupts the transcription process. Nevertheless, tigecycline is not affected by common tetracycline resistance mechanisms, such as acquired *tetA* genes. However, the *tetX3* and *tetX4* genes (capable of degrading all tetracyclines) have been reported to confer high-level acquired tigecycline resistance [44]. Moreover, tigecycline resistance in *E. coli* is frequently the result of the constitutive expression of resistance–nodulation–cell division (RND)-type efflux pumps [45]. One of the main structures is the tripartite efflux system *AcrAB–TolC*, which allows the efflux of multiple antibiotics. Upregulation of this system apparently leads to tigecycline resistance in *Enterobacterales* [46]. The *acrA* and *acrB* genes are located in an identical operon regulated by the local repressor *acrR*. Additionally, global regulators such as *mar*A are involved in *E. coli* MDR regulation. This gene is regulated by the repressors *marR* and *acrB*, which are activated in the absence of *marR* [47].

All *K. pneumoniae* isolates in this study presented reduced susceptibility to TGC. Mutations in *ramR* G25S are possibly related to MIC = 4, K194* to MIC = 2, and S157P to MIC = 8 (Appendix A). Within *K. pneumoniae*, *ramA* acts as an *AraC/XylS* transcriptional activator that controls the *AcrAB–TolC* efflux pump, and *ramA* expression is downregulated by *ramR*. Therefore, mutations in *ramR* can cause chromosomal resistance to tigecycline [48]. Additionally, genetic and phenotypic analysis of the *E. coli* isolates suggested that this decreased susceptibility to tigecycline within the *E. coli* isolates could be related to missense mutations in *marA* and frameshift mutation across *acrR* genes. This study presents a variety of mutations linked to phenotypic resistance to colistin in *K. pneumoniae*, together with resistance to tigecycline in *E. coli* and *K. pneumoniae*. However, functional validation is still required to establish their contributions to such resistance levels.

Regarding the genetic environment of *bla*_KPC_, all *K. pneumoniae* strains analyzed in this study harbored *bla*_KPC-2_ and *bla*_KPC-3_ alleles inside a ~10 kb Tn*4401* transposon, with the genetic context tnpR–tnpA–istA–istB–bla_KPC_–tnpA. To date, two basic structures have been described as carrying *bla*_KPC_ genes—the *Tn4401* transposon and the Tn*3–*Tn*4401* chimera—both found to be linked to its successful mobilization [49]. However, in South America, *bla*_KPC_ genes are predominantly linked to Tn*4401* [50]. In this study, the identification of *bla*_KPC_ genes inside of a Tn*4401* module in *K. pneumoniae* suggests that these strains are reservoirs and could disseminate carbapenem resistance in clinical settings. Conversely, only one isolate for *E. coli* presented a carbapenemase resistance gene. However, this strain was sensitive to carbapenems, and the genetic environment for *bla*_KPC-3_ could not be established. Therefore, we believe this gene to be non-functional.

*bla*_TEM_ genes, especially *bla*_TEM-1_, were identified in most of the isolates. However, these genes encode lactamases that are not able to inactivate 3GCs. Therefore, it is not practical to estimate the contribution of these enzymes to the overall resistance observed in the strains producing CTX-M or KPC enzymes. The same rationale is considered for other *bla*_TEM_ variants, *bla*_SHV_, and *bla*_OXA_ lactamases, although some variants may be from the ESBL group, like those registered in this genomic description.

The genomic epidemiology of *mcr*-1 genes in Latin America needs to be enhanced. At first glance, plasmids belonging to the IncI2 incompatibility group are an important genetic backbone for *mcr*-1 gene mobilization in our setting. In Latin America, IncI2-like plasmids have been reported in Chile, Argentina, Venezuela, Bolivia, and Uruguay [51,52,53]. However, the data need to be enhanced with more local studies in the region to reveal their epidemiological implications. The *mcr*-1 gene’s epidemiology is an example of this. Furthermore, the same issue arises with all the genes and genetic structures reported in this study [54].

### Limitations

The isolates were selected for their resistance to colistin, encompassing both *E. coli* and *K. pneumoniae*. These represent the full spectrum of resistant isolates collected, thanks to collaboration between various groups in Quito, Ecuador. It was only possible to include *K. pneumoniae* isolates from human clinical environments. Additionally, these isolates were not addressed in terms of prevalence or outbreaks. Our aim was to identify the mechanisms of resistance to colistin and associated resistances, focusing on phenotypic profiles through genomic data.

Our motivation for sharing this work was to contribute data on the molecular epidemiology of antibiotic resistance in Ecuador, a context where available information is limited. Our study falls within the “One Health” approach, which emphasizes the interdependence among human, animal, and environmental health. A relevant finding is the identification of the *mcr*-1 gene in a common plasmid genetic environment among isolates from humans, animals, and the environment, reinforcing this approach. A global epidemiological map of bacterial resistance is valuable to the extent that it reaches a level of detail that allows for the identification of relationships between ecological, evolutionary, and clinical aspects. Each finding equally contributes to the understanding of genomic epidemiology, as genetic determinants spread among interconnected bacterial communities across the biome, sharing an inherent potential for dissemination, accelerated by current human community connectivity.

The scientific method is built around hypotheses that can be confirmed or refuted, and the methodology adopted must be capable of addressing these questions. Therefore, the inherent limitations of the techniques employed should not be seen as criteria for questioning the validity of the results obtained. Every technology presents limitations, but these do not undermine its robustness in resolving the questions posed when considering its strengths and weaknesses. If we were to only consider the limitations of each method, no methodology would be useful. In this context, the application of short-read massive sequencing adopted in this study has been established as a fundamental tool in microbiology at the genomic level. It is worth noting that this technique is critical for sequencing the human genome, which attests to its reliability.

Our study did not establish new breakpoints but relied on these pre-existing interpretations to ensure consistency with the original analyses. The primary goal of our work was to confirm the resistance profiles reported by the contributing institutions and validate the isolates for whole-genome sequencing. Consequently, our analysis focused on identifying and characterizing the genetic elements associated with resistance, as described in this manuscript.

The isolates characterized through whole-genome sequencing in this report were initially isolated in separate studies. These isolates were selected based on the criteria established by the respective research groups that donated them. Consequently, this introduces inherent biases related to each study’s phenotypic screening cut-offs. Given the isolates’ diverse origins—spanning environmental, veterinary, and human clinical contexts—specific criteria were employed in accordance with the distinct focus of each field. Therefore, we recommend that prevalence data be interpreted with consideration of these criteria and the specific conditions of the prior studies. Nevertheless, it is important to note that for the selection of extended-spectrum β-lactamase (ESBL)-producing isolates, all studies consistently employed the double-disk synergy test, as recommended by CLSI methodology.

In our case, we utilized the VITEK AST card to reconfirm ESBL production for the specific purpose of selecting isolates that carried the genetic elements of interest for whole-genome sequencing. This approach ensured that we focused on strains containing the genes we aimed to characterize and describe in detail in this study.

The susceptibility testing covered all of our available resources. Therefore, antibiotics such as ceftazidime–avibactam were not tested. However, in the specific case of this therapeutic combination, the absence of metallo-carbapenemases or avibactam-insensitive *bla*_KPC_-type carbapenemases provides valuable clues about the sensitivity of the isolates through genomic analysis. It is worth noting that the *E. coli* isolates were sensitive to carbapenems, indicating an option currently available for our community.

## 4. Materials and Methods

### 4.1. Initial Characterization of the Isolates

The isolates were initially characterized in the scope of different studies. In brief, this study considered nine 3GC-resistant *E. coli* isolates harboring the *mcr*-1 gene (six from poultry farms, two from human infections, and one from the compost of a dairy farm), together with ten colistin/carbapenem-resistant *K. pneumoniae*, isolated from clinical samples:All isolates from poultry were identified in 2014 in a previous study conducted by our group [22].The clinical isolates were selected from a 2016 sample collection of 4000 *Enterobacteriaceae* gathered by Zurita&Zurita Laboratories.The compost isolate was identified during an analysis of the prevalence of *bla*_CTX-M_ in an agricultural productive unit in Quito, Ecuador, conducted in 2016 (unpublished data).

### 4.2. Sample Collection and Preparation for Sequencing

The isolates, obtained from three independent projects, had their characteristics—such as colistin resistance, carbapenem resistance, the presence of the *mcr*-1 gene, and ESBL production—identified and used as inclusion criteria in the present study. Upon completion of these projects, the isolates were stored in cryovials at ultralow temperatures for subsequent genome sequencing and phenotypic analysis, which were confirmed in this study. Prior to sequencing, the isolates were reactivated by inoculation in a liquid medium. Single colonies were isolated on selective media: TBX medium (BIO-RAD, Hercules, CA, USA) supplemented with cefotaxime (3 µg/mL) for *E. coli* and MacConkey medium (Difco, Detroit, MI, USA) supplemented with imipenem (1 µg/mL) for *K. pneumoniae.* A single colony was selected for phenotypic analysis and cultured in LB broth, from which DNA was extracted using a commercial column-based total DNA extraction kit. The obtained DNA was used for PCR and sequencing purposes.

### 4.3. Antibiotic Susceptibility Testing

Antimicrobial susceptibility testing for colistin (COL) was established using the microdilution method employing GNX2F plates (Thermo Scientific™, West Palm Beach, FL, USA), while the minimum inhibitory concentration (MIC) breakpoint was 2 (EUCAST). The presence of the mcr-1 gene was established using the PCR method [20]. MIC values for ampicillin (AMP), piperacillin–tazobactam (TPZ), cefoxitin (FOX), ceftazidime (CAZ), ceftriaxone (CRO), cefepime (FEP), doripenem (DOR), ertapenem (ERT), imipenem (IMI), meropenem (MEM), amikacin (AK), gentamicin (GEN), ciprofloxacin (CIP), and tigecycline (TIG) were all obtained through the Vitek^®^ 2 system via the AST-N272 card (Biomérieux™, Marcy-l’Étoile, France). This card additionally allowed the identification of ESBL production. The results were evaluated with breakpoints and recommendations from the CLSI guide [52]. Tigecycline MIC interpretation was carried out using the EUCAST 2019 clinical breakpoint for *E. coli* and ECOFF for *K. pneumoniae*.

### 4.4. Whole-Genome Sequencing

The whole-genome sequencing of 19 isolates was performed using the MiSeq^®^ platform (Illumina, San Diego, CA, USA). Genomic libraries were prepared using the NexteraXT^®^ kit (Illumina, San Diego, CA, USA). Raw data were processed using the CLC Genomics™ WorkBench^®^ tool, version 8.5. First, a quality filter was applied to the reads, excising regions of low quality and possible contamination with adapters, followed by discarding reads with <25 nucleotides. The quality-filtered reads were used to perform de novo assemblies for each isolate.

### 4.5. Mating Assay and PCR-Based Replicon Typing (PBRT)

The mating assay and PBRT were carried out using the UNIETAR laboratory protocol. In brief, the *E coli* J53 strain, resistant to sodium azide (200 µg/mL), was used as a receptor. Mating was carried out in LB broth by mixing every *E. coli* isolate and the same receptor strain, followed by incubation for 18 h at 35 °C. The transconjugants were selected from LB agar plates supplemented with sodium azide (200 µg/mL) and colistin (2 µg/mL). The transconjugants were evaluated using colony PCR for *mcr*-1 gene 20 and PBRT using the PBRT 2.0 kit (Diatheva, Fano, Italy).

### 4.6. Plasmid Analysis

PlasmidFinder v.2.1, along with the map-to-reference tool available in Genious Prime software R10.0 and the cgview_comparison_tool (https://github.com/paulstothard/cgview_comparison_tool, accessed on 19 October 2023), was used for plasmid reconstruction.

### 4.7. Data Management and Statistical Analysis

Whole-genome sequencing of the isolates formed part of the Bioproject “Colistin-resistant *E. coli* and *K. pneumoniae* from Ecuador”, conducted at Universidad del Bosque, Bogotá, Colombia.

The identification of species was carried out using the Strain Seeker^®^ program [53]. Annotation of each genome was performed using the RAST^®^ tool [54]. Sequences are available from NCBI under accession number PRJNA507384.

Phylogenetic analysis was carried out using Center for Genomic Epidemiology (CGE) tools: NDtree 1.2 (NDtree constructs phylogenetic trees from single-end or pair-end reads), TreeViewer 1.0, and MLST tool (multi-locus sequence typing from an assembled genome or from a set of reads) using default settings, as well as VirulenceFinder (identification of acquired virulence genes) and SeroTypeFinder (prediction of serotypes in total or partial sequenced isolates of *E. coli*). These tools are available at https://www.genomicepidemiology.org/, accessed on 19 October 2023. An enterobase-based SNPtree was used for the MLST *E. coli* analysis.

Resistance genes were identified through the ResFinder^®^ service (CGE), and integrons were identified using IntFinder (https://github.com/kalilamali/Integrons, accessed on 19 October 2023). The sequences of the laboratory standard strain *E. coli* K-12 (GenBank accession, NC_000913.3) and the ATCC strain of *K. pneumoniae* ATCC BAA-2146 (GenBank accession: NZ_CP006659.2) were used as references for putative mutation discovery using Genious Prime software. Genetic environment reconstruction was carried out using the map-to-reference tool.

## 5. Conclusions

This study investigated nine 3GC-resistant *E. coli* isolates carrying the *mcr-1* gene (sourced from poultry farms, human infections, and dairy farm compost), alongside ten clinical isolates of colistin- and carbapenem-resistant *K. pneumoniae*. The human-origin *E. coli* isolates were classified as ST609 (phylogroup A), while those from the poultry and compost represented phylogroups A, B1, E, and F. The diverse *K. pneumoniae* sequence types included ST13, ST258, and ST86. The key resistance genes identified in *E. coli* included *bla*_CTX-M-55_, *bla*_CTX-M-65_, *bla*_CTX-M-15_, and *bla*_CTX-M-2_, with *bla*_CMY-2_ and *bla*_KPC-3_ also detected. All *mcr-*1.1 genes were plasmid-borne within IncI2 plasmids. Reduced tigecycline susceptibility in *E. coli* was linked to *marA* and *acrA* gene mutations.

In *K. pneumoniae*, 3GC resistance was associated with *bla*_CTX-M-15_ and *bla*_CTX-M-65_, while carbapenem resistance was mediated by *blaKPC-2* and *bla*_KPC-3_. Colistin resistance correlated with genetic alterations in *mgrB*, *marA*, *acrA*, *arnB*, *eptA*, *pmrB*, *pmrJ*, and *phoQ*, and tigecycline resistance with *ramR* mutations. This study also identified integron structures, with Int191(*dfrA14*) being the most prevalent across isolates. These findings enhance our understanding of multidrug-resistant *E. coli* and *K. pneumoniae* within the genomic epidemiology context, supporting the global One Health framework.

## Figures and Tables

**Figure 1 antibiotics-14-00206-f001:**
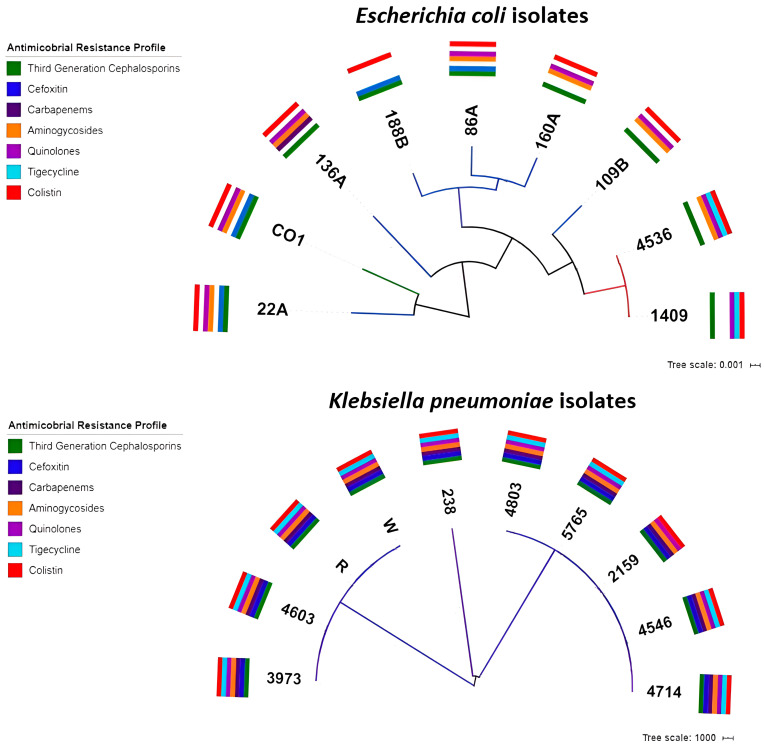
**SNP tree determined for the *E. coli* and *K. pneumoniae* isolates.** Colored bars indicate the antibiotic resistance profile of each isolate.

**Figure 2 antibiotics-14-00206-f002:**
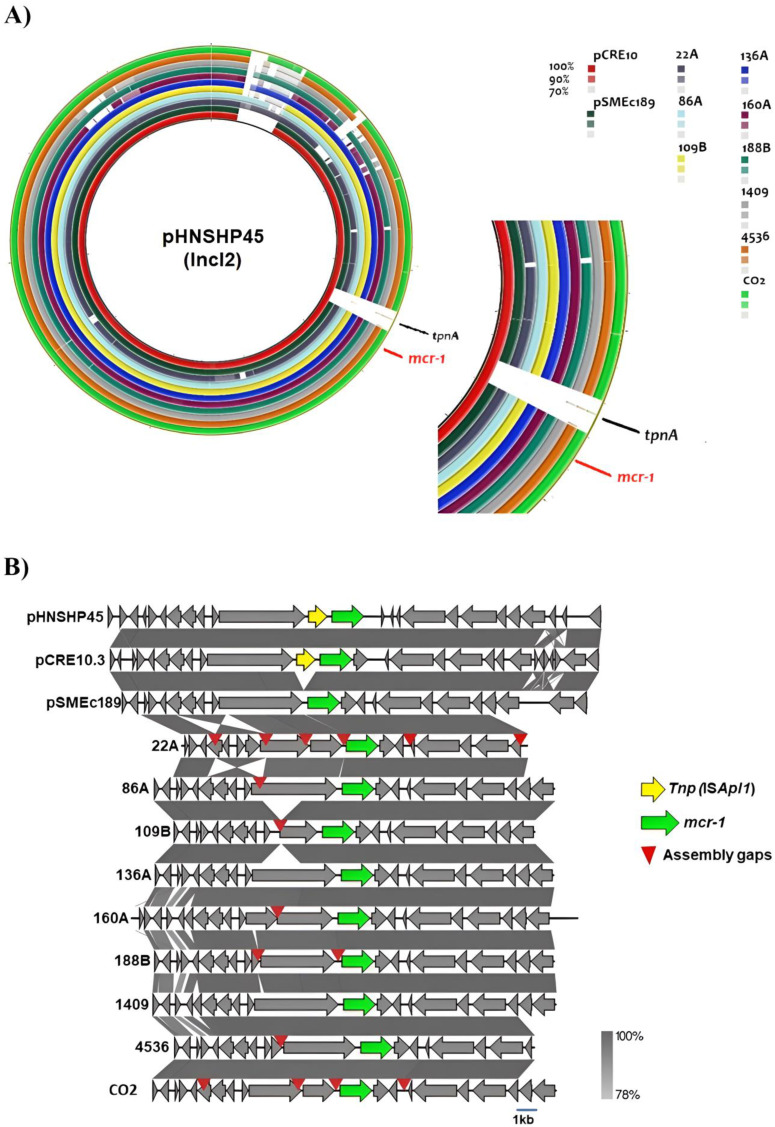
Plasmid reconstruction and alignment. Alignment of *mcr*-1 harboring IncI2 plasmid references pHNSHP45, pCRE10, and pSMEc189 with reconstructed plasmids from *E. coli* isolates harboring *mcr*-1 genes was performed. The zoomed view shows the *mcr-*1 neighborhood. Graphic alignment of the *mcr*-1.1 genetic context (**A**). IS*Apl1* transposase and *mcr-*1 genes are highlighted. Red arrowheads represent gaps between contigs in the draft plasmids (**B**).

**Figure 3 antibiotics-14-00206-f003:**
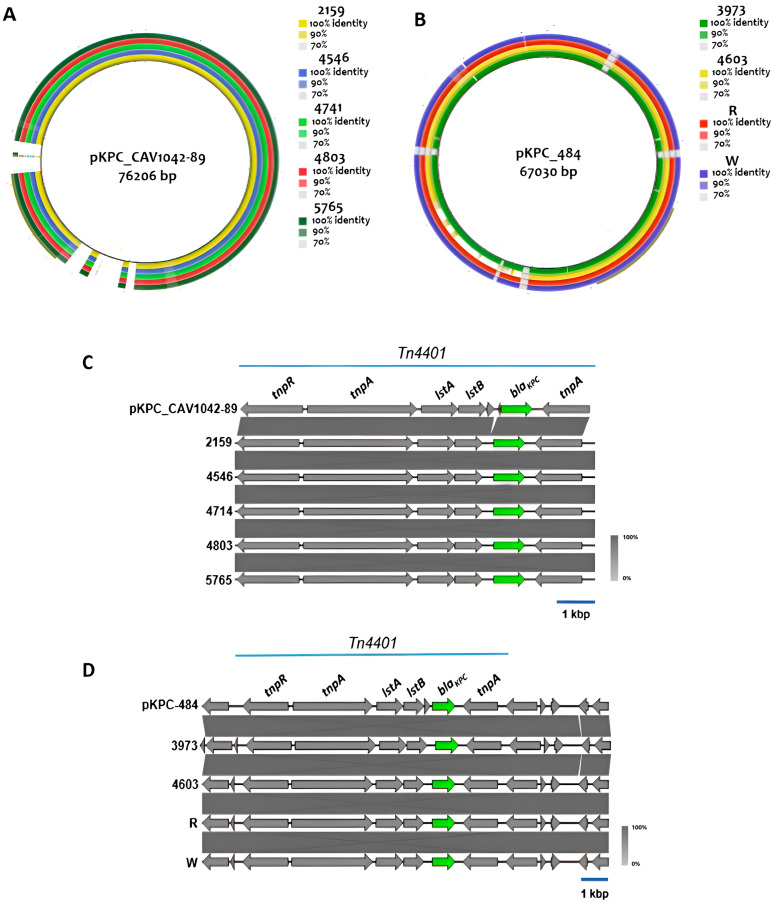
Ring alignment of *K. pneumoniae* ST13 isolate draft genomes against the previously described plasmid pKPC-CAV1042-89 (**A**) and ST258 strain daft genomes against pKPC-484 (**B**). Alignment of Tn*4401* loci from ST13 isolates with the pKPC-CAV1042-89 reference (**C**) and from ST258 strains with the pKPC-484 reference (**D**).

**Table 1 antibiotics-14-00206-t001:** Antibiotic resistance profiles.

Isolate	Species	Date	Origin	Age	Location	AMP	TPZ	FOX	CAZ	CRO	FEP	DOR	ERT	IMI	MEM	AK	GEN	CIP	TIG	COL
1409	*E. coli*	10 February 2016	Peritoneal liquid	14	Quito	**16**	≤4	8	**16**	**≥64**	2	≤0.12	≤0.5	≤0.25	≤0.25	≤2	≤1	**≥4**	**2**	**4**
4536	*E. coli*	10 June 2016	Wound secretion	72	Quito	**32**	64	≤4	**16**	**≥64**	2	≤0.12	≤0.5	≤0.25	≤0.25	8	**≥16**	**≥4**	1	**4**
CO2 (CO1)	*E. coli*	2016	Compost dairy farm	-	Quito	**≥32**	≤4	**16**	**16**	**≥64**	**≥64**	≤0.12	≤0.5	≤0.25	≤0.25	≤2	**≥16**	**≥4**	≤0.5	**4**
86A	*E. coli*	2014	Poultry	-	Yaruquí	**≥32**	≤4	**16**	4	**≥64**	**≥64**	≤0.12	≤0.5	≤0.25	≤0.25	≤2	**≥16**	**≥4**	≤0.5	**8**
109B	*E. coli*	2014	Poultry	-	El Chota	**≥32**	≤4	8	≤1	**≥64**	2	≤0.12	≤0.5	≤0.25	≤0.25	≤2	**≥16**	**≥4**	≤0.5	**4**
136A	*E. coli*	2014	Poultry	-	Santo Domingo	**16**	≤4	≤4	≤1	**16**	≤1	≤0.12	≤0.5	≤0.25	≤0.25	≤2	**≥16**	**≥4**	≤0.5	**8**
22A	*E. coli*	2014	Poultry	-	Ascazubi	**≥32**	**≥128**	**≥64**	**16**	**16**	≤1	≤0.12	≤0.5	≤0.25	≤0.25	≤2	**≥16**	**≥4**	≤0.5	**8**
160A	*E. coli*	2014	Poultry	-	Guayllabamba	**≥32**	≤4	8	**32**	**≥64**	**4**	0.5	≤0.5	≤0.25	≤0.25	4	**≥16**	**≥4**	≤0.5	**≥16**
188B	*E. coli*	2014	Poultry	-	Ibarra	**16**	≤4	**≥64**	4	**8**	≤1	≤0.12	≤0.5	≤0.25	≤0.25	≤2	≤1	≤0.25	≤0.5	**8**
1220672 (W)	*K. pneumoniae*	12 February 2016	Urine	61	Quito	**≥32**	**≥128**	**≥64**	**≥64**	**≥64**	**≥64**	**≥8**	**≥8**	**≥16**	**≥16**	**≥64**	**≥16**	**≥4**	**4**	**≥16**
1204191 (R)	*K. pneumoniae*	12 February 2016	Blood	52	Quito	**≥32**	**≥128**	**≥64**	**≥64**	**≥64**	**≥64**	**≥8**	**≥8**	**≥16**	**≥16**	**≥64**	**≥16**	**≥4**	**4**	**≥16**
238	*K. pneumoniae*	2 June 2016	Cervical abscess	54	Quito	**16**	≤4	≤4	≤1	≤1	≤1	≤0.12	≤0.5	≤0.25	≤0.25	≤2	≤1	≤0.5	2	**4**
2159	*K. pneumoniae*	3 March 2016	Wound secretion	37	Quito	**≥32**	**≥128**	**32**	**≥64**	**≥64**	**≥64**	**≥8**	**≥8**	**≥16**	**≥16**	**16**	**≥16**	**2**	2	**≥16**
3973	*K. pneumoniae*	19 May 2016	Tracheal secretion	55	Quito	**≥32**	**≥128**	**≥64**	**≥64**	**≥64**	**≥64**	**≥8**	**≥8**	**≥16**	**≥16**	**≥64**	**≥16**	**≥4**	**4**	**≥16**
4546	*K. pneumoniae*	10 June 2016	Surgical wound discharge	56	Quito	**≥32**	**≥128**	**≥64**	**≥64**	**≥64**	**≥64**	**≥8**	**≥8**	**8**	**≥16**	**≥64**	**≥16**	**≥4**	2	**≥16**
4603	*K. pneumoniae*	13 June 2016	Tracheal secretion	64	Quito	**≥32**	**≥128**	**≥64**	**≥64**	**≥64**	**≥64**	**≥8**	**≥8**	**≥16**	**≥16**	**≥64**	**≥16**	**≥4**	**≥8**	**≥16**
4714	*K. pneumoniae*	17 June 2016	Blood	52	Quito	**≥32**	**≥128**	**≥64**	**≥64**	**≥64**	**≥64**	**≥8**	**≥8**	**8**	**≥16**	**≥64**	**≥16**	**≥4**	2	**≥16**
4803	*K. pneumoniae*	21 June 2016	Tracheal secretion	95	Quito	**≥32**	**≥128**	**≥64**	**≥64**	**≥64**	**≥64**	**≥8**	**≥8**	**≥16**	**≥16**	**≥64**	**≥16**	**≥4**	2	**≥16**
5765	*K. pneumoniae*	26 July 2016	Sputum	71	Quito	**≥32**	**≥128**	**≥64**	**≥64**	**≥64**	**≥64**	**≥8**	**≥8**	**8**	**≥16**	**≥64**	**≥16**	**≥4**	2	**≥16**

Resistance profiles for colistin-resistant *E. coli* and *K. pneumoniae* isolates. Ampicillin (AMP), piperacillin–tazobactam (TPZ), cefoxitin (FOX), ceftazidime (CAZ), ceftriaxone (CRO), cefepime (FEP), doripenem (DOR), ertapenem (ERT), imipenem (IMI), meropenem (MEM), amikacin (AK), gentamicin (GEN), ciprofloxacin (CIP), tigecycline (TIG), and colistin (COL). Resistance MIC data are in bold. Tigecycline MIC interpretation was carried out using the EUCAST 2019 clinical breakpoint for *E. coli* and ECOFF for *K. pneumoniae*.

**Table 2 antibiotics-14-00206-t002:** Serotypes and virulence factors in the *E. coli* isolates.

Isolate	Serotype	*gad*	*iss*	*mchF*	*cma*	*lpfA*	*iroN*	*air*	*eilA*	*iha*	*celB*	*ccl*	*ireA*	*cba*	*astA*	Total per Isolate
**1409 ͳ**	O9:H4	X	X													2
**4536 ͳ**	O9:H4	X	X								X					3
**CO2 α**	H25	X	X	X	X	X		X	X							7
**86A ***	0115:H28	X	X		X	X									X	5
**109B ***	02:H4		X	X	X	X							X			5
**136A ***	H28	X	X	X	X		X	X	X	X		X		X		10
**22A ***	H34	X		X				X	X	X						5
**160A ***	0157:H21	X	X	X		X	X						X			6
**188A ***	H20	X				X										2
**Total per gene**		8	7	5	4	4	2	3	3	2	1	1	1	1	1	

**ͳ**, human isolates; **α**, compost isolate; *****, poultry isolates; *gad,* glutamate decarboxylase; *iss,* increased serum survival; *mchF*, ABC transporter protein MchF; *cma*, Colicin M; *lpfA*, long polar fimbriae; *iroN*, enterobactin siderophore receptor protein; *air*, enteroaggregative immunoglobulin repeat protein; *eilA*, salmonella *hilA* homolog; *iha*, adherence protein; *celB*, endonuclease colicin E2; *ccl*, cloacin; *ireA*, siderophore receptor; *cba*, colicin B; *astA*, EAST-1 heat-stable toxin.

## Data Availability

Publicly available datasets were analyzed in this study. This data can be found here: [https://www.ncbi.nlm.nih.gov] [PRJNA507384]. accessed on 15 January 2023.

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
