# Peer review of "Genomic Insights into Colistin and Tigecycline Resistance in ESBL-Producing *Escherichia coli* and *Klebsiella pneumoniae* Harboring *bla*_KPC_ Genes in Ecuador"

_antibiotics, 2025, doi:10.3390/antibiotics14020206_

Round 1
Reviewer 1 Report
Comments and Suggestions for Authors
The authors identified genomic structure alteration in Escherichia coli and Klebsiella pneumoniae resistant to third-generation cephalosporins. The strains were isolated from humans, poultry, and a dairy- farm environment within Ecuador. They used a bioinformatic methods for the strain characterization and for genomic changes identification associated to colistin and tigecycline resistance in the both species. The study was well-designed and carefully conducted. However, I have a few comments for the authors to address to improve the manuscript:
Describe in more detail how the samples for sequencing were obtained. For example, the DNA isolation was obtained from a single colony, the bacterial culture conditions, the storage conditions, etc.
Please, describe the program used for phylogenetic analysis as well as the thresholds adopted to construct the phylogenetic tree. In Figure 1, includes the STs of the samples and describe the colors of the branches, thus facilitating the interpretation of the tree, in addition to correlating it with the text.
The first paragraph of page 4 is repeated with the previous paragraph.
Revise the various typing errors, for example: in the Abstract: “phylogroups A, B1, E, and F. K and diverse STs. K. pneumoniae isolates”, the allele appears in bold; In the Results: E. coli is not in italic font; Footnote of Table 1; in the Discussion in paragraphs 2 and 3. In the second line of page 26, the citation of ref. 53 is missing.
In the supplementary data, please insert a title and a short text to understand the content of the table.
Author Response
- Describe in more detail how the samples for sequencing were obtained. For example, the DNA isolation was obtained from a single colony, the bacterial culture conditions, the storage conditions, etc.
Sample Collection and Preparation for Sequencing
The isolates were obtained from three independent projects, in which their characteristics—such as colistin resistance, carbapenem resistance, presence of the mcr-1 gene, and ESBL) production—were identified, and used as inclusion criteria in the present study. Upon completion of these projects, the isolates were stored in cryovials at ultralow temperatures for subsequent genome sequencing and phenotypic analyses confirmation to this study. Prior to sequencing, the isolates were reactivated by inoculation in liquid medium. Single colonies were isolated on selective media: TBX (3µg/mL) medium supplemented with cefotaxime for E. coli and MacConkey medium supplemented with imipenem (1µg/mL) for K. pneumoniae. A single colony was selected for phenotypic analysis and cultured in LB broth, from which DNA was extracted using a commercial column based total DNA extraction kit, the obtained DNA was used for PCR and sequencing purposes.
- Please, describe the program used for phylogenetic analysis as well as the thresholds adopted to construct the phylogenetic tree. In Figure 1, includes the STs of the samples and describe the colors of the branches, thus facilitating the interpretation of the tree, in addition to correlating it with the text.
Response Description was added and figure was enhanced.
The phylogenetic analysis was carried out using the Center for Genomic Epidemiology (CGE) tools. NDtree 1.2 (NDtree constructs phylogenetic trees from Single-End or Pair-End), TreeViewer 1.0 and MLST tool (Multi Locus Sequence Typing from an assembled genome or from a set of reads), using default settings. As well as VirulenceFinder (Identifcation of acquired virulence genes) and SeroTypeFinder (Prediction of serotypes in total or partial sequenced isolates of E. coli). These tools are available at https://www.genomicepidemiology.org/.
- The first paragraph of page 4 is repeated with the previous paragraph.
Response: The paragraph was improved
- Revise the various typing errors, for example: in the Abstract: “phylogroups A, B1, E, and F. K and diverse STs. pneumoniaeisolates”, the allele appears in bold; In the Results: E. coli is not in italic font; Footnote of Table 1; in the Discussion in paragraphs 2 and 3. In the second line of page 26, the citation of ref. 53 is missing.
Response: The typing was improved following the recommendations.
- In the supplementary data, please insert a title and a short text to understand the content of the table.
Response: A title and a short text was added.
This supplementary table contains the detailed data of genetic determinants and their relation with the antibiotics tested for each isolate.
Reviewer 2 Report
Comments and Suggestions for Authors
A good paper that merits publication.
I have recently had double infection from this mix of bacteria and it is not pleasant to have or to be treated for.
The figures are above average quality and aid greatly.
Publish after minor English language correction.
Author Response
- The figures are above average quality and aid greatly
Response:
The quality and resolution of the figures were enhanced using the tools available: canva, paint and word.
Publish after minor English language correction.
English language was improved
Reviewer 3 Report
Comments and Suggestions for Authors
Dear Authors,
You presented an original work submitted by Zurita J. et al. entitled: Genomic insights into colistin and tigecycline resistance in ESBL- producing Escherichia coli and Klebsiella pneumoniae harboring blaKPC genes in Ecuador.
Your manuscript is relevant since it presents a genomic characterization for E. coli and K. pneumoniae resistant to last-line antibiotics.
However, major revisions are suggested before the publication of the manuscript
1) This study focuses on the genomic characterization of resistant strains of E. coli and K. pneumoniae harboring the mcr-1 gene. Due to the relevance of these pathogenic bacterial strains, a more forceful introduction is suggested to reflect the importance of addressing these type of public health problems.
2) Only 19 strains of resistant bacteria were used in this work. Can the authors indicate whether this population is statistically representative?
3) Since the genomic characterization was not sufficiently conclusive, a more proactive discussion on Table 1 is needed.
4) I kindly suggest that the authors include a geographical analysis comparing their results with others published for the South American region.
5) Relevant bibliography must be added and discussed in your manuscript: DOI: 10.3390/antibiotics12030488 ; DOI: 10.1038/s41467-018-03205-z; https://iris.who.int/bitstream/handle/10665/376776/9789240093461-eng.pdf
6) 18% (10/55) self-citation is too much in my opinion. I kindly suggest looking for alternative references if is this possible.
7) Some typos must be corrected
Line 20 …resistance to third-generation cephalosporins (3GCs).. instead (3CG)
Line 44 Keywords were missing
8) Some references include DOI and others do not. Homogenize the format of references
Reference 25 must be corrected
Regards
Author Response
- This study focuses on the genomic characterization of resistant strains of E. coli and K. pneumoniae harboring the mcr-1 gene. Due to the relevance of these pathogenic bacterial strains, a more forceful introduction is suggested to reflect the importance of addressing these type of public health problems.
Response: Introduction was reorganized and improved
- Introduction
In recent years, the emergence of Enterobacterales strains that are resistant to all β-lactam antibiotics has restricted therapeutic options to only last-line antimicrobials, including colistin and tigecycline [1,2]. Furthermore, the emergence of Escherichia coli (E. coli) and Klebsiella pneumoniae (K. pneumonjiae) strains that are resistant to multiple antibiotics, including third-generation cephalosporins (3GCs), carbapenems, colistin, and tigecycline, has markedly affected the ability to treat patients, particularly within developing countries [3].
The introduction of 3GCs as a response for increasing antimicrobial resistance to classic antibiotics (first generations of β-lactams, tetracyclines, quinolones, folate pathway inhibitors, and sulfonamides), provided a degree of hope for the for treating complex infections driven by Enterobacterales. However, the extended use of 3GCs to treat human and animal infections compromised their utility, mostly due to the emergence of extended-spectrum β-lactamase (ESBL) genes [4,5]. Consequently, in response to the dissemination of ESBL genes blaTEM, blaSHV, and blaCTX-M among Enterobacterales, carbapenems became the antibiotic golden standard for treating multi-drug resistant (MDR) bacteria. Unfortunately, during this last decade, the acquisition of genes coding for enzymes capable of hydrolyzing carbapenems (e.g., blaKPC and blaNDM, among others) has become an urgent public health issue, jeopardizing the treatment of severe infections caused by Enterobacterales [5].
The lack of reliable therapeutic options to treat carbapenem-resistant Enterobacterales (CRE) led to utilizing older compounds, such as polymyxins. Indeed, colistin (polymyxin E), was employed in veterinary medicine for decades, mainly for the prophylaxis and treatment of infections due to Enterobacterales, and was also employed as a growth-promoting agent in selected countries [6,7]. Since polymyxins class were employed against CRE, resistance gradually emerged (mostly due to mutations within chromosomal genes) [8], becoming a major public health priority, particularly within developing nations [9]. Furthermore, the report of mobile elements carrying colistin resistance determinants (e.g., mobile colistin resistance, mcr genes) was a highly concerning development [10]. Notably, availability of novel antimicrobial agents (e.g., β-lactam/β-lactamase inhibitors) with potent activity against CRE, has markedly decreased the use of polymyxins within developed nations.
In 2005, tigecycline was approved by the Food and Drug Administration (FDA) for the treatment of skin / soft-tissue and intra-abdominal infections [11]. Due to tigecycline’s in vitro activity against CRE, this antibiotic is typically employed as salvage therapy for CRE [12] (mostly in combination with other agents), and resistance in Enterobacterales remains minimal. Nonetheless, resistance to tigecycline has been characterized as involving mutations in ribosomal proteins, modifications within efflux pumps, and the presence of tet(A) and tet(M), among others [13]. Recently, the presence of the mobile determinant tetX4, coding for a tetracycline-inactivating enzyme, was described as driving tigecycline resistance within animal isolates [14]. In Ecuador alone, blaCTX-M, blaKPC and mcr-1 genes in Escherichia coli have been reported within isolates from humans, food, poultry, pigs, dog feces, and the environment [15-26]. However, this issue needs research and community communication strategies [27]. Furthermore, carbapenem-resistant K. pneumoniae has been described in human infections within Ecuador since 2010 [28] and colistin-resistant K. pneumoniae is reportable through the Ecuadorian national surveillance program and reported in the One Health approach [29]. However, the genomic epidemiology of Enterobacterales resistant to colistin and last-resort antibiotics, together with potential determinants of resistance, remain to be elucidated.
Consequently, the aim of this study was to perform genomic characterizations for E. coli and K. pneumoniae within human, poultry and environmental isolates that were resistant to last-line antibiotics within Quito-Ecuador.
- Only 19 strains of resistant bacteria were used in this work. Can the authors indicate whether this population is statistically representative?
Isolates initial characterization
Isolates were initially characterized in the scope of differing studies. In brief, this study considered nine 3CG-resistant Escherichia coli isolates harboring the mcr-1 gene (six from poultry farms, two from human infections, and one from the compost of a dairy-farm), together with ten colistin- / carbapenem-resistant Klebsiella pneumoniae, isolated from clinical samples.
- All isolates from poultry were identified in 2014 within a previous study conducted by our group [52].
- Clinical isolates were selected from a 2016 Zurita&Zurita Laboratories-conducted collection of 4,000 Enterobacteriaceae sample gathering exercise.
- Compost isolate was identified during the analysis of the prevalence of blaCTX-M in an agricultural productive unit in Quito, Ecuador, conducted in 2016. Unpublished data.
The isolates analyzed were the total of isolates of E. coli-producing mcr-1 and carbapemen- and colistin-resistant K. pneumoniae available at our location to our study group.
Our motivation for sharing this work has been to contribute data on the molecular epidemiology of antibiotic resistance in Ecuador, a context where available information is limited. Our study falls within the "One Health" approach, which emphasizes the interdependence between human, animal, and environmental health. A relevant finding is the identification of the mcr-1 gene in a common plasmid genetic environment among isolates from humans, animals, and the environment, reinforcing this approach. The global epidemiological map of bacterial resistance is valuable to the extent that it reaches a level of detail that allows for the identification of relationships between ecological, evolutionary, and clinical aspects. Each finding equally contributes to the understanding of genomic epidemiology, as genetic determinants spread among interconnected bacterial communities across the biome, sharing an inherent potential for dissemination, accelerated by current human community connectivity.
- Since the genomic characterization was not sufficiently conclusive, a more proactive discussion on Table 1 is needed.
Response:
Our motivation for sharing this work has been to contribute data on the molecular epidemiology of antibiotic resistance in Ecuador, a context where available information is limited. Our study falls within the "One Health" approach, which emphasizes the interdependence between human, animal, and environmental health. A relevant finding is the identification of the mcr-1 gene in a common plasmid genetic environment among isolates from humans, animals, and the environment, reinforcing this approach. The global epidemiological map of bacterial resistance is valuable to the extent that it reaches a level of detail that allows for the identification of relationships between ecological, evolutionary, and clinical aspects. Each finding equally contributes to the understanding of genomic epidemiology, as genetic determinants spread among interconnected bacterial communities across the biome, sharing an inherent potential for dissemination, accelerated by current human community connectivity.
4) I kindly suggest that the authors include a geographical analysis comparing their results with others published for the South American region.
5) Relevant bibliography must be added and discussed in your manuscript: DOI: 10.3390/antibiotics12030488 ; DOI: 10.1038/s41467-018-03205-z; https://iris.who.int/bitstream/handle/10665/376776/9789240093461-eng.pdf
Response: The cites were added in a comprehensive paragraph
Genomic epidemiology of mcr-1 genes in Latin America needs to be enhanced. At first glance, plasmid belong to the IncI2 incompatibility group are an important genetic backbone for mcr-1 gene mobilization at our setting. In Latin America, the IncI2-like plasmids have been reported in Chile, Argentina, Venezuela, Bolivia and Uruguay [52-54]. However, the data needs to be enhanced with more local studies in the region to reveal its epidemiological implications. The gene mcr-1 epidemiology is an example of this need. Furthermore, the same issue is shared with all the genetic genes and structures reported in this study [55].
6) 18% (10/55) self-citation is too much in my opinion. I kindly suggest looking for alternative references if is this possible.
Response: We includes all of our publications because represent the epidemiology reveled before 10 years working in this issue at the our location. Therefore are relevant to depict our epidemiology and support the need of the genomic characterization at our location that additionally are interconnected to the worldwide epidemiology under an One Health approach.
7) Some typos must be corrected
Line 20 …resistance to third-generation cephalosporins (3GCs).. instead (3CG)
Line 44 Keywords were missing
8) Some references include DOI and others do not. Homogenize the format of references
Reference 25 must be corrected
Response: All suggestions were implemented
Reviewer 4 Report
Comments and Suggestions for Authors
1. I would like to express my gratitude for the opportunity to review the manuscript entitled “Genomic insights into colistin and tigecycline resistance in ESBL- producing Escherichia coli and Klebsiella pneumoniae harboring blaKPC genes in Ecuador” However, there are some major concerns that must be addressed in order to be considered for publication.
2. I noticed that the manuscript does not include a section for keywords, which is essential for indexing and improving discoverability. Please ensure to add this section in your revised version.
3. I recommend consolidating the introduction section by combining paragraphs with similar content, as the current structure consists of seven short paragraphs. This will enhance clarity and coherence, aligning better with the principles of effective research article introductions.
4. I noticed that the section on Materials and Methods lacks a dedicated subsection for "Data Management and Statistical Analysis". It would be beneficial to include this information to enhance the transparency and reproducibility of the study.
5. I have concerns regarding the sample selection methods and the limited spatio-temporal scope of the research, which may affect the robustness of the epidemiological analysis. Could you please provide more details about the duration of the study to help address these potential limitations in diversity and sample bias?
6. Including specific details about the disposable VITEK AST card in the manuscript would enhance transparency and reproducibility, allowing readers to understand the methodology and potentially replicate the study.
7. I noticed that the antimicrobial resistance profiles in Table 1 include samples from both animal and human sources, analyzed using the Vitek 2® system. Could you please clarify whether the MIC breakpoints used in the Vitek system are based on human or veterinary standards?
8. Based on your findings, the presence of the blaCTX-M-15, blaCTX-M-65, blaKPC-2, and blaKPC-3 genes is significant. However, it would be prudent to assess the presence of other ESBL genes, such as blaTEM, blaSHV, and blaOXA-1, for comprehensive confirmation.
9. The methodology for antimicrobial susceptibility testing and whole-genome sequencing is well-detailed; however, I noticed that a phenotypic confirmation assay for ESBL producers has not been performed. This step is crucial for accurately identifying and confirming the presence of ESBL-producing isolates, which could significantly impact the study's findings. Could you clarify the rationale behind omitting this important assay?
10. I recommend improving the quality and resolution of the figures, as several, particularly Figure 1, Figure 2A, Figure 3A, and Figure 3B, are difficult to read and interpret. Enhancing these visuals would significantly aid in conveying the study's results more clearly.
11. It appears that the manuscript combines “Concluding Remarks and Limitations” in the same section. In my opinion, these should be separated for clarity. The summary section must be both concise and comprehensive, clearly stating the key findings and offering specific conclusions that provide valuable insights into clinical applications.
12. The references cited in this manuscript provide support for the statements made; however, I recommend including additional recent references to ensure the information remains up to date. Currently, the manuscript cites only 12.72% (7/55) of the most recent publications within the past five years, including four publications in 2020, two in 2021, and one in 2024.
13. I noticed inconsistency in the use of the terms "antimicrobial" and "antibiotic" throughout the manuscript, as seen in phrases like "last-line antibiotics and last-line antimicrobials" and "antibiotic resistance and antimicrobial resistance." It is crucial to clarify which terminology is most appropriate for your manuscript to ensure consistency and accuracy in your writing.
14. I suggest that when introducing a new species, the full genus name should be written out initially, followed by the abbreviated form in subsequent mentions. Additionally, please verify the accuracy of bacterial names throughout the manuscript to ensure scientific correctness.
Author Response
-
- Comment: I would like to express my gratitude for the opportunity to review the manuscript entitled “Genomic insights into colistin and tigecycline resistance in ESBL- producing Escherichia coli and Klebsiella pneumoniae harboring blaKPC genes in Ecuador” However, there are some major concerns that must be addressed in order to be considered for publication.
- Comment: I noticed that the manuscript does not include a section for keywords, which is essential for indexing and improving discoverability. Please ensure to add this section in your revised version.
Response: Keywords was added
Keywords: Carbapenem-resistance, colistin, Ecuador, Escherichia coli, Klebsiella pneumoniae, KPC, mcr-1, tigecycline.
- I recommend consolidating the introduction section by combining paragraphs with similar content, as the current structure consists of seven short paragraphs. This will enhance clarity and coherence, aligning better with the principles of effective research article introductions.
Response: the introduction section was improved
- Introduction
In recent years, the emergence of Enterobacterales strains that are resistant to all β-lactam antibiotics has restricted therapeutic options to only last-line antimicrobials, including colistin and tigecycline [1,2]. Furthermore, the emergence of Escherichia coli (E. coli) and Klebsiella pneumoniae (K. pneumonjiae) strains that are resistant to multiple antibiotics, including third-generation cephalosporins (3GCs), carbapenems, colistin, and tigecycline, has markedly affected the ability to treat patients, particularly within developing countries [3].
The introduction of 3GCs as a response for increasing antimicrobial resistance to classic antibiotics (first generations of β-lactams, tetracyclines, quinolones, folate pathway inhibitors, and sulfonamides), provided a degree of hope for the for treating complex infections driven by Enterobacterales. However, the extended use of 3GCs to treat human and animal infections compromised their utility, mostly due to the emergence of extended-spectrum β-lactamase (ESBL) genes [4,5]. Consequently, in response to the dissemination of ESBL genes blaTEM, blaSHV, and blaCTX-M among Enterobacterales, carbapenems became the antibiotic golden standard for treating multi-drug resistant (MDR) bacteria. Unfortunately, during this last decade, the acquisition of genes coding for enzymes capable of hydrolyzing carbapenems (e.g., blaKPC and blaNDM, among others) has become an urgent public health issue, jeopardizing the treatment of severe infections caused by Enterobacterales [5].
The lack of reliable therapeutic options to treat carbapenem-resistant Enterobacterales (CRE) led to utilizing older compounds, such as polymyxins. Indeed, colistin (polymyxin E), was employed in veterinary medicine for decades, mainly for the prophylaxis and treatment of infections due to Enterobacterales, and was also employed as a growth-promoting agent in selected countries [6,7]. Since polymyxins class were employed against CRE, resistance gradually emerged (mostly due to mutations within chromosomal genes) [8], becoming a major public health priority, particularly within developing nations [9]. Furthermore, the report of mobile elements carrying colistin resistance determinants (e.g., mobile colistin resistance, mcr genes) was a highly concerning development [10]. Notably, availability of novel antimicrobial agents (e.g., β-lactam/β-lactamase inhibitors) with potent activity against CRE, has markedly decreased the use of polymyxins within developed nations.
In 2005, tigecycline was approved by the Food and Drug Administration (FDA) for the treatment of skin / soft-tissue and intra-abdominal infections [11]. Due to tigecycline’s in vitro activity against CRE, this antibiotic is typically employed as salvage therapy for CRE [12] (mostly in combination with other agents), and resistance in Enterobacterales remains minimal. Nonetheless, resistance to tigecycline has been characterized as involving mutations in ribosomal proteins, modifications within efflux pumps, and the presence of tet(A) and tet(M), among others [13]. Recently, the presence of the mobile determinant tetX4, coding for a tetracycline-inactivating enzyme, was described as driving tigecycline resistance within animal isolates [14]. In Ecuador alone, blaCTX-M, blaKPC and mcr-1 genes in Escherichia coli have been reported within isolates from humans, food, poultry, pigs, dog feces, and the environment [15-25]. However, this issue needs research and community communication strategies [26]. Furthermore, carbapenem-resistant K. pneumoniae has been described in human infections within Ecuador since 2010 [27] and colistin-resistant K. pneumoniae is reportable through the Ecuadorian national surveillance program and reported in the One Health approach [28]. However, the genomic epidemiology of Enterobacterales resistant to colistin and last-resort antibiotics, together with potential determinants of resistance, remain to be elucidated.
Consequently, the aim of this study was to perform genomic characterizations for E. coli and K. pneumoniae within human, poultry and environmental isolates that were resistant to last-line antibiotics within Quito-Ecuador.
- I noticed that the section on Materials and Methods lacks a dedicated subsection for "Data Management and Statistical Analysis". It would be beneficial to include this information to enhance the transparency and reproducibility of the study.
Response: A Data Management and Statistical Analysis subsection was added
Data Management and Statistical Analysis
Whole-genome sequencing of isolates formed part of the Bioproject: Colistin-resistant Escherichia coli and Klebsiella pneumoniae from Ecuador, conducted at Universidad del Bosque. Bogotá, Colombia.
Identification of species was carried out using the Strain Seeker® program [54]. Annotation of each genome was performed using the RAST® tool [55]. Sequences are available at NCBI, under the accession number PRJNA507384.
Resistance genes were identified through the Resfinder® service and Integrons by Intfinder (Center for Genomic Information). The chromosome sequence for colistin-sensitive Klebsiella pneumoniae (ATCC BAA-2146; NZ_CP006659.2) was used as a reference for mgrB, papP, phoQ, phoP, pmrA, pmrB, eptA, eptB lipid A modification genes, together with arnA_DH/FT, arnB, arnC, arnT, pmrJ, pmrL, phoB lipid A Ara4N pathway genes, and analyzed using Geneious Prime® software package.
Putative mutations for reduced susceptibility to tigecycline in Klebsiella pneumoniae and Escherichia co6li. The chromosome sequence of tigecycline-sensitive Escherichia coli K-12 (GenBank accession: NC_000913.3) was used as a reference for marR and acrR genes. Chromosome sequence of colistin sensitive Klebsiella pneumoniae ATCC BAA-2146 (GenBank accession: NZ_CP006659.2) was used as a reference for ramR, acrR and oqxR.
- I have concerns regarding the sample selection methods and the limited spatio-temporal scope of the research, which may affect the robustness of the epidemiological analysis. Could you please provide more details about the duration of the study to help address these potential limitations in diversity and sample bias?
Response: The details about the duration of the study have been added
Isolates initial characterization
Isolates were initially characterized in the scope of differing studies. In brief, this study considered nine 3CG-resistant Escherichia coli isolates harboring the mcr-1 gene (six from poultry farms, two from human infections, and one from the compost of a dairy-farm), together with ten colistin- / carbapenem-resistant Klebsiella pneumoniae, isolated from clinical samples.
- All isolates from poultry were identified in 2014 within a previous study conducted by our group [52].
- Clinical isolates were selected from a 2016 Zurita&Zurita Laboratories-conducted collection of 4,000 Enterobacteriaceae sample gathering exercise.
- Compost isolate was identified during the analysis of the prevalence of blaCTX-M in an agricultural productive unit in Quito, Ecuador, conducted in 2016. Unpublished data.
The isolates analyzed were the total of isolates of E. coli-producing mcr-1 and carbapemen- and colistin-resistant K. pneumoniae available at our location to our study group.
- Including specific details about the disposable VITEK AST card in the manuscript would enhance transparency and reproducibility, allowing readers to understand the methodology and potentially replicate the study.
Response: The details about the phenotypic screening have been improved.
Antimicrobial susceptibility testing
Antimicrobial susceptibility testing MIC values for colistin (COL) were established by micro-dilution, using GNX2F plates (Thermo Scientific™, West Palm Beach, USA), using the minimum inhibitory concentration (MIC ≥ 2) breackpoint, and the precence of mcr-1 gene was stablished by pcr [52]. MIC values for ampicillin (AMP), piperacillin-tazobactam (TPZ), cefoxitin (FOX), ceftazidime (CAZ), ceftriaxone (CRO), cefepime (FEP), doripenem (DOR), ertapenem (ERT), imipenem (IMI), meropenem (MEM), amikacin (AK), gentamicin (GEN), ciprofloxacin (CIP) and tigecycline (TIG), were all obtained through the Vitek® 2 system, via AST-N272 card (Biomérieux™, Marcy-l'Étoile, France). This card additionally allows the identification of ESBL production. Results were evaluated with breakpoints and recommendations from the CLSI guide [53].
- I noticed that the antimicrobial resistance profiles in Table 1 include samples from both animal and human sources, analyzed using the Vitek 2® system. Could you please clarify whether the MIC breakpoints used in the Vitek system are based on human or veterinary standards?
Response: The VITEK AST card results were analyzed using CLSI breakpoints (human)
MIC values for ampicillin (AMP), piperacillin-tazobactam (TPZ), cefoxitin (FOX), ceftazidime (CAZ), ceftriaxone (CRO), cefepime (FEP), doripenem (DOR), ertapenem (ERT), imipenem (IMI), meropenem (MEM), amikacin (AK), gentamicin (GEN), ciprofloxacin (CIP) and tigecycline (TIG), were all obtained through the Vitek® 2 system, via AST-N272 card (Biomérieux™, Marcy-l'Étoile, France). This card additionally allows the identification of ESBL production. Results were evaluated with breakpoints and recommendations from the CLSI guide
- Based on your findings, the presence of the blaCTX-M-15, blaCTX-M-65, blaKPC-2, and blaKPC-3 genes is significant. However, it would be prudent to assess the presence of other ESBL genes, such as blaTEM, blaSHV, and blaOXA-1, for comprehensive confirmation.
Response: The details about the blaTEM, blaSHV, and blaOXA-1 have been added
The blaTEM genes, especially blaTEM-1 was identified in most of the isolates. However, these genes codify to lactamases that are not able to inactivate 3GC. Therefore, its not practical the estimation of the contribution of these enzymes to the overall resistance observed in the strains producing CTX-M or KPC enzymes. The same rational is considered to other blaTEM variants, blaSHV and blaOXA lactamases. although some variants may be from the ESBL group, like the registered in this genomic description.
- The methodology for antimicrobial susceptibility testing and whole-genome sequencing is well-detailed; however, I noticed that a phenotypic confirmation assay for ESBL producers has not been performed. This step is crucial for accurately identifying and confirming the presence of ESBL-producing isolates, which could significantly impact the study's findings. Could you clarify the rationale behind omitting this important assay?
Response: The VITEK AST card include the phenotypic ESBL confirmation.
Antimicrobial susceptibility testing
Antimicrobial susceptibility testing MIC values for colistin (COL) were established by micro-dilution, using GNX2F plates (Thermo Scientific™, West Palm Beach, USA), using the minimum inhibitory concentration (MIC ≥ 2) breackpoint, and the precence of mcr-1 gene was stablished by pcr [52]. MIC values for ampicillin (AMP), piperacillin-tazobactam (TPZ), cefoxitin (FOX), ceftazidime (CAZ), ceftriaxone (CRO), cefepime (FEP), doripenem (DOR), ertapenem (ERT), imipenem (IMI), meropenem (MEM), amikacin (AK), gentamicin (GEN), ciprofloxacin (CIP) and tigecycline (TIG), were all obtained through the Vitek® 2 system, via AST-N272 card (Biomérieux™, Marcy-l'Étoile, France). This card additionally allows the identification of ESBL production. Results were evaluated with breakpoints and recommendations from the CLSI guide [53]. Tigecycline MIC interpretation was carried out using EUCAST 2019 clinical breakpoint for Escherichia coli and ECOFF for Klebsiella pneumoniae.
- I recommend improving the quality and resolution of the figures, as several, particularly Figure 1, Figure 2A, Figure 3A, and Figure 3B, are difficult to read and interpret. Enhancing these visuals would significantly aid in conveying the study's results more clearly.
Response: The quality and resolution of the figures were enhanced using the tools available: canva, paint and word.
- It appears that the manuscript combines “Concluding Remarks and Limitations” in the same section. In my opinion, these should be separated for clarity. The summary section must be both concise and comprehensive, clearly stating the key findings and offering specific conclusions that provide valuable insights into clinical applications.
Response: The Limitations section has been separated.
Limitations
The isolates were selected for their resistance to colistin, encompassing both E. coli and K. pneumoniae. These represent …………………………………………………. available for our community.
- The references cited in this manuscript provide support for the statements made; however, I recommend including additional recent references to ensure the information remains up to date. Currently, the manuscript cites only 12.72%(7/55) of the most recent publications within the past five years, including four publications in 2020, two in 2021, and one in 2024.
Response: The references were up dated.
- I noticed inconsistency in the use of the terms "antimicrobial" and "antibiotic" throughout the manuscript, as seen in phrases like "last-line antibiotics and last-line antimicrobials" and "antibiotic resistance and antimicrobial resistance." It is crucial to clarify which terminology is most appropriate for your manuscript to ensure consistency and accuracy in your writing.
Response: Inconsistencies in the use of the terms "antimicrobial" and "antibiotic" were revised.
- I suggest that when introducing a new species, the full genus name should be written out initially, followed by the abbreviated form in subsequent mentions. Additionally, please verify the accuracy of bacterial names throughout the manuscript to ensure scientific correctness.
Response: The full genus name was improved when a new species is mentioned for the first time, followed by the abbreviated.
Abstract: Escherichia coli (E. coli) and Klebsiella pneumoniae (K. pneumoniae) resistant to third-generation
- Introduction
In recent years, the emergence of Enterobacterales strains that are resistant to all β-lactam antibiotics has restricted therapeutic options to only last-line antimicrobials, including colistin and tigecycline [1,2]. Furthermore, the emergence of Escherichia coli (E. coli) and Klebsiella pneumoniae (K. pneumonjiae) strains
Reviewer 5 Report
Comments and Suggestions for Authors
This manuscript describes E. coli and K. pneumoniae isolates of different origins in Ecuador resistant to last-line antibiotics (3CG, carbapenems, tigecycline, and colistin). The mechanisms responsible for these resistance phenotypes were tested and reported, as well as the ST typing of the isolates. The significance of this type of data is of great importance on a global scale, but the manuscript needs to be thoroughly revised as some parts are missing in this version of the manuscript. My revision is partial, as there are some gaps in the manuscript, and I will send the complete version as soon as I read the version corrected by the authors.
Abstract
Line 20 – The sentence „Mainly 20 within the developing world” should not be separated sentences.
Line 22 – characterization of
Line 24 – the part at genomic level is redundant since in previous sentence it was explained that characterization is on genomic level
Line 28 - F. K and diverse STs, the letter K is probably some mistake
Line 29 – isolates should not be italic
Line 31 - blaCMY-2 and blaKPC-3 (the latter in a carbapenem-susceptible isolate) in E. coli.
Line 32 - (reference for assembling 32 CP018669) – this part is redundant for abstract
Line 35 - carbapenem resistance, respectively; Tn4401
Line 36 – tnpR-tnpA-istA-istB-blaKPC-2-tnpA should not be bold
All genes in this section except bla genes should be written in italics in complete, eg. mgrB instead of mgrB
Lines 36-39 – it should be indicated in which species (E. coli or K. pneumoniae) these mutations were found
Lines 38-39 – nonsense and missense
Line 39 – integron
Line 41 – Int
Lines 41-43 – this sentence should be rephrased
Line 44 – the keywords are missing
Introduction
Line 56 – for is doubled in this sentence
Line 61 – became the gold standard; multidrug-resistant
Line 72 - developing countries
Line 74 – this part of the sentence “was a highly concerning development” is unclear and should be modified
Line 76 - developing countries; this sentence needs a reference
Line 78 – soft tissue
Line 82 – it is better to write alternation or substitutions of proteins than mutations
Line 83 – tet(A) is the efflux pump
Lines 84-86 – this sentence should be part of previous paragraph since it is associated to tigecycline resistance; in which species it was found?
Line 86 – E. coli
Line 88 – carbapenem-resistant
Line 89 – this part of the sentence “within Ecuador since” is unclear and should be modified
Line 91 – colistin is the last-resort antibiotic
Line 92 - remains
Line 93 – characterization of
The sections Material and Methods and Results should be organized in subsections.
Material and Methods
This section was without lines, so my corrections are numbered:
1. Enterobacteriaceae should be in italics
2. E. coli, K. pneumoniae – shorter version
3. , Bogota
4. The identification number of projects or reference should be included
5. mcr-1 should be in italics
6. This sentence “Results were evaluated with breakpoints and recommendations from the [53]” should be corrected. It should be indicated from the which source. The sentence is incomplete.
7. ResFinder is a service on the website of the Center for Genomic Epidemiology, not Information, while Intfinder could not be found on the same website, so the source of this service should be included.
8. All genes in this section except bla genes should be written in italics in complete, eg. mgrB instead of mgrB.
9. K. pneumoniae ATCC BAA-2146 (GenBank accession: NZ_CP006659.2)
10. Correct this Escherichia co6li to E. coli
11. This sentence “Chromosome sequence of colistin sensitive Klebsiella pneumoniae ATCC BAA-2146 (GenBank accession: NZ_CP006659.2) was used as a reference for ramR, acrR and oqxR” should be moved in previous paragraph were the K. pneumoniae was mentioned. If the intention was to separate the genes responsible for colistin and tigecycline resistance, that should be explained. In first paragraph colistin, and in the second tigecycline, but this part “Chromosome sequence of colistin sensitive Klebsiella pneumoniae ATCC BAA-2146” should be changed to tigecycline-sensitive.
Results
Line 97 - This sentence “Antibiotic susceptibility profile Table 1 depicts results for susceptibility analyses” should be corrected to “Antibiotic susceptibility profiles depicts results for susceptibility analyses (Table 1)”.
Line 100 - all, excluding one isolate, were resistant to ciprofloxacin
Lines 105-107 – The sentence “Determinants of antibiotic resistance Escherichia coli Whole Genome Sequencing (WGS) alloy to identify acquired genes and mutations related to phenotypic antibiotic resistance” is not well organized. Probably, the authors wanted to write “Whole Genome Sequencing (WGS) of the tested isolates allow identification of acquired determinants of antibiotic resistance and mutations related to phenotypic antibiotic resistance”.
Line 107 – in supplementary table has been written blaCTX-M-14, so it should be checked where is the mistake; in this table were also included blaTEM and this gene was not mentioned in the text
Lines 110-111 – This sentence “Tigecycline-resistant - or with reduced susceptibility to these antibiotics - harbored missense mutations within efflux system regulatory genes marA and acrA” should be rephrased e.g. “The isolates resistant or with reduced susceptibility to tigecycline…”
Line 112 – Why blaTEM and blaSHV were not included in the text like ESBLs?
Line 113 – serin carbapenemases
Supplementary Data – this document should be checked in the parts that should be in italics or not
Lines 123-124 - ST13 (n = 5), ST258 (n = 4), and ST86(n = 1)
Line 124 – E. coli should be in italics
Line 127 – The numbers of ST - ST258, ST238, and ST13 – are not correct
Line 129 – This part “Serotypes and virulence of Escherichia coli isolates” is probably left by mistake
In Supplementary the serotypes should be changed like they are in the text e.g. O9:H4
Is compost E. coli CO1 or CO2? The name is not the same in the manuscript. Also, in supplementary there are some addition in the names (1 CT).
Line 131 - , respectively – this is redundant in this sentence
Line 132 – each isolate harbors 2 -10 genes, not 1 -8; it should be indicated which gene was the most prevalent and which isolates harbored the most virulence genes (10 of them)
Line 133 – this sentence is completely unspecified. What is classic E. coli pathotypes? It should be explained, but this explanation is more suitable for discussion.
Lines 134-138 – this part is duplicated
Line 142 – bp
The tools used for the results obtained in Figures 1, 2 and 3, and Table 2 must be included and explained in Material and methods. This is completely missed.
Line 145 – which mcr-1 primers? This also were not included in Material and methods section. Protocol of conjugation was not included as well.
Table 2.
1. there is a mistake in the number of genes for isolate 86A (5 instead of 6); lpf vertically 5 instead of 4; iroN 2 instead of 4
2. the names of the genes should be in italics; air instead of Air
3. the symbols for the isolates of different source in the legend are not visible
4. in the legend the names of the genes should be written in the same style like the rest of the text and explained the same for each gene e.g. iroN, enterobactin siderophore receptor protein
Table 1.
1. Cervical instead of cervical
2. MIC values for GEN for two isolates (1409,188B) are confusing, probably it should be ≤ 1
2. MEM (meropenem) instead of MER; in the legend meropenem (MEM)
3. For tigecycline MIC was used Eucast document (2019) for E. coli, but it should be included reference. Information for K. pneumoniae is even more incomplete.
Comments on the Quality of English LanguageThe English could be improved to more clearly express the research.
Author Response
Abstract
Line 20 – The sentence „Mainly 20 within the developing world” should not be separated sentences.
Corrected
Line 22 – characterization of
corrected
Line 24 – the part at genomic level is redundant since in previous sentence it was explained that characterization is on genomic level
corrected
Line 28 - F. K and diverse STs, the letter K is probably some mistake
correted
Line 29 – isolates should not be italic
corrected
Line 31 - blaCMY-2 and blaKPC-3 (the latter in a carbapenem-susceptible isolate) in E. coli.
corrected
Line 32 - (reference for assembling 32 CP018669) – this part is redundant for abstract
corrected
Line 35 - carbapenem resistance, respectively; Tn4401
corrected
Line 36 – tnpR-tnpA-istA-istB-blaKPC-2-tnpA should not be bold
All genes in this section except bla genes should be written in italics in complete, eg. mgrB instead of mgrB
corrected
Lines 36-39 – it should be indicated in which species (E. coli or K. pneumoniae) these mutations were found
added
Lines 38-39 – nonsense and missense
corrected
Line 39 – integron
corrected
Line 41 – Int
corrected
Lines 41-43 – this sentence should be rephrased
Rephrased
Line 44 – the keywords are missing
added
Introduction
Line 56 – for is doubled in this sentence
corrected
Line 61 – became the gold standard; multidrug-resistant
corrected
Line 72 - developing countries
corrected
Line 74 – this part of the sentence “was a highly concerning development” is unclear and should be modified
modified
Line 76 - developing countries; this sentence needs a reference
removed
Line 78 – soft tissue
corrected
Line 82 – it is better to write alternation or substitutions of proteins than mutations
corrected
Line 83 – tet(A) is the efflux pump
corrected
Lines 84-86 – this sentence should be part of previous paragraph since it is associated to tigecycline resistance; in which species it was found?
reorganized
Line 86 – E. coli
corrected
Line 88 – carbapenem-resistant
corrected
Line 89 – this part of the sentence “within Ecuador since” is unclear and should be modified
modified
Line 91 – colistin is the last-resort antibiotic
corrected
Line 92 - remains
corrected
Line 93 – characterization of
corrected
The sections Material and Methods and Results should be organized in subsections.
Material and Methods
This section was without lines, so my corrections are numbered:
- Enterobacteriaceae should be in italics
revised
- coli, K. pneumoniae – shorter version
revised
- , Bogota
corrected
- The identification number of projects or reference should be included
Included when possible
- mcr-1 should be in italics
implemented
- This sentence “Results were evaluated with breakpoints and recommendations from the [53]” should be corrected. It should be indicated from the which source. The sentence is incomplete.
corrected
- ResFinder is a service on the website of the Center for Genomic Epidemiology, not Information, while Intfinder could not be found on the same website, so the source of this service should be included.
https://github.com/kalilamali/Integrons was added
- All genes in this section except bla genes should be written in italics in complete, eg. mgrB instead of mgr
Revised and improved
- pneumoniaeATCC BAA-2146 (GenBank accession: NZ_CP006659.2)
I did not understand the change suggested
- Correct this Escherichia co6li to coli
corrected
- This sentence “Chromosome sequence of colistin sensitive Klebsiella pneumoniae ATCC BAA-2146 (GenBank accession: NZ_CP006659.2) was used as a reference for ramR, acrR and oqxR” should be moved in previous paragraph were the pneumoniaewas mentioned. If the intention was to separate the genes responsible for colistin and tigecycline resistance, that should be explained. In first paragraph colistin, and in the second tigecycline, but this part “Chromosome sequence of colistin sensitive Klebsiella pneumoniae ATCC BAA-2146” should be changed to tigecycline-sensitive.
corrected
Results
Line 97 - This sentence “Antibiotic susceptibility profile Table 1 depicts results for susceptibility analyses” should be corrected to “Antibiotic susceptibility profiles depicts results for susceptibility analyses (Table 1)”.
corrected
Line 100 - all, excluding one isolate, were resistant to ciprofloxacin
Corrected
Lines 105-107 – The sentence “Determinants of antibiotic resistance Escherichia coli Whole Genome Sequencing (WGS) alloy to identify acquired genes and mutations related to phenotypic antibiotic resistance” is not well organized. Probably, the authors wanted to write “Whole Genome Sequencing (WGS) of the tested isolates allow identification of acquired determinants of antibiotic resistance and mutations related to phenotypic antibiotic resistance
Changed
Line 107 – in supplementary table has been written blaCTX-M-14, so it should be checked where is the mistake; in this table were also included blaTEM and this gene was not mentioned in the text
Corrected and added
The blaTEM genes, especially blaTEM-1 was identified in most of the isolates. However, these genes codify to lactamases that are not able to inactivate 3GC. Therefore, its not practical the estimation of the contribution of these enzymes to the overall resistance observed in the strains producing CTX-M or KPC enzymes. The same rational is considered to other blaTEM variants, blaSHV and blaOXA lactamases. although some variants may be from the ESBL group, like the registered in this genomic description.
Lines 110-111 – This sentence “Tigecycline-resistant - or with reduced susceptibility to these antibiotics - harbored missense mutations within efflux system regulatory genes marA and acrA” should be rephrased e.g. “The isolates resistant or with reduced susceptibility to tigecycline…”
changed
Line 112 – Why blaTEM and blaSHV were not included in the text like ESBLs?
Included
The blaTEM genes, especially blaTEM-1 was identified in most of the isolates. However, these genes codify to lactamases that are not able to inactivate 3GC. Therefore, its not practical the estimation of the contribution of these enzymes to the overall resistance observed in the strains producing CTX-M or KPC enzymes. The same rational is considered to other blaTEM variants, blaSHV and blaOXA lactamases. although some variants may be from the ESBL group, like the registered in this genomic description.
Line 113 – serin carbapenemases
Supplementary Data – this document should be checked in the parts that should be in italics or not
revised
Lines 123-124 - ST13 (n = 5), ST258 (n = 4), and ST86(n = 1)
corrected
Line 124 – E. coli should be in italics
corrected
Line 127 – The numbers of ST - ST258, ST238, and ST13 – are not correct
corrected
Line 129 – This part “Serotypes and virulence of Escherichia coli isolates” is probably left by mistake
Revised
In Supplementary the serotypes should be changed like they are in the text e.g. O9:H4
Is compost E. coli CO1 or CO2? The name is not the same in the manuscript. Also, in supplementary there are some addition in the names (1 CT).
Revised
Line 131 - , respectively – this is redundant in this sentence
removed
Line 132 – each isolate harbors 2 -10 genes, not 1 -8; it should be indicated which gene was the most prevalent and which isolates harbored the most virulence genes (10 of them)
corrected
Line 133 – this sentence is completely unspecified. What is classic E. coli pathotypes? It should be explained, but this explanation is more suitable for discussion.
corrected
intestinal or extra-intestinal E. coli pathotypes.
Lines 134-138 – this part is duplicated
removed
Line 142 – bp
corrected
The tools used for the results obtained in Figures 1, 2 and 3, and Table 2 must be included and explained in Material and methods. This is completely missed.
The explanation was added
Whole-genome sequencing of isolates formed part of the Bioproject: Colistin-resistant E. coli and K. pneumoniae from Ecuador, conducted at Universidad del Bosque. Bogota, Colombia.
Identification of species was carried out using the Strain Seeker® program [54]. Annotation of each genome was performed using the RAST® tool [55]. Sequences are available at NCBI, under the accession number PRJNA507384.
The phylogenetic analysis was carried out using the Center for Genomic Epidemiology (CGE) tools. NDtree 1.2 (NDtree constructs phylogenetic trees from Single-End or Pair-End), TreeViewer 1.0 and MLST tool (Multi Locus Sequence Typing from an assembled genome or from a set of reads), using default settings. As well as VirulenceFinder (Identifcation of acquired virulence genes) and SeroTypeFinder (Prediction of serotypes in total or partial sequenced isolates of E. coli). These tools are available at https://www.genomicepidemiology.org/.
Resistance genes were identified through the Resfinder® service (CGE) and integros were identified by Intfinder (https://github.com/kalilamali/Integrons). The sequence of laboratory standard strain E. coli K-12 (GenBank accession: NC_000913.3) and ATCC strain of K. pneumoniae ATCC BAA-2146 (GenBank accession: NZ_CP006659.2) were used as references for putative mutations discovering using Genious Prime software. As well as for genetic environments reconstruction using the tool map to reference.
Line 145 – which mcr-1 primers? This also were not included in Material and methods section. Protocol of conjugation was not included as well.
Table 2.
- there is a mistake in the number of genes for isolate 86A (5 instead of 6); lpfvertically 5 instead of 4; iroN2 instead of 4
- the names of the genes should be in italics; air instead of Air
- the symbols for the isolates of different source in the legend are not visible
- in the legend the names of the genes should be written in the same style like the rest of the text and explained the same for each gene e.g. iroN, enterobactin siderophore receptor protein
Cite and the laboratory of protocol details repository were added
All of the E. coli J53 transconjugants selected for colistin resistance produced amplicons with mcr-1-specific primers [20], and none of them produced amplicons following PCR-based replicon typing (PBRT). The mating assay and PBRT were carried out under the UNIETAR laboratory protocol.
Table 1.
- Cervical instead of cervical
- MIC values for GEN for two isolates (1409,188B) are confusing, probably it should be ≤ 1
corrected
- MEM (meropenem) instead of MER; in the legend meropenem (MEM)
Corrected
- For tigecycline MIC was used Eucast document (2019) for E. coli, but it should be included reference. Information for K. pneumoniae is even more incomplete.
Added
Antimicrobial susceptibility testing
Antimicrobial susceptibility testing MIC values for colistin (COL) were established by micro-dilution, using GNX2F plates (Thermo Scientific™, West Palm Beach, USA), using the minimum inhibitory concentration (MIC ≥ 2) breackpoint, and the precence of mcr-1 gene was stablished by pcr [52]. MIC values for ampicillin (AMP), piperacillin-tazobactam (TPZ), cefoxitin (FOX), ceftazidime (CAZ), ceftriaxone (CRO), cefepime (FEP), doripenem (DOR), ertapenem (ERT), imipenem (IMI), meropenem (MEM), amikacin (AK), gentamicin (GEN), ciprofloxacin (CIP) and tigecycline (TIG), were all obtained through the Vitek® 2 system, via AST-N272 card (Biomérieux™, Marcy-l'Étoile, France). This card additionally allows the identification of ESBL production. Results were evaluated with breakpoints and recommendations from the CLSI guide [53]. Tigecycline MIC interpretation was carried out using EUCAST 2019 clinical breakpoint for E. coli and ECOFF for K. pneumoniae.
Round 2
Reviewer 3 Report
Comments and Suggestions for Authors
Dear Authors,
Since all comments were satisfactorily resolved, in my opinion the article can be accepted in its present form for publication.
Regards
Author Response
Dear Reviewer,
Thank for your kind help to improve the manuscript
Reviewer 4 Report
Comments and Suggestions for Authors#7 Thank you for your clarification regarding the use of CLSI breakpoints for the VITEK AST card results. However, I would like to emphasize that there are indeed different breakpoints for antibiotics between bacteria isolated from human and animal sources. Could you please discuss how this may impact the interpretation of your resistance profiles, particularly given that your samples include both human and animal-derived isolates?
#9 Thank you for your response regarding the VITEK AST card's capability for phenotypic ESBL confirmation. While I appreciate this point, I would like to reiterate that the gold standard for ESBL detection involves specific confirmatory tests that are essential for accurately identifying ESBL producers. Methods such as the Double Disk Synergy Test or Etest provide additional validation beyond screening methods. Additionally, the CLSI guidelines specify that ESBL testing should include Ceftazidime (0.25-128 µg/mL), Ceftazidime-clavulanate (0.25/4-128/4 µg/mL), Cefotaxime (0.25-64 µg/mL), and Cefotaxime-clavulanate (0.25/4-64/4 µg/mL). Testing necessitates using both cefotaxime and ceftazidime, both alone and in combination with clavulanate.
#11 I appreciate that you addressed my previous comment by separating the Limitations section. However, I noticed that the Conclusion section is still missing from the manuscript. It is important to include a distinct Conclusion section that summarizes the key findings and offers insights into clinical applications.
#12 Regarding the updating of references, I still note a total of 55 references, and it is unclear which ones have been updated. Could you please specify which references have been revised to include more recent publications?
#13 Thank you for addressing the inconsistencies in the use of the terms "antimicrobial" and "antibiotic." However, I still noticed some discrepancies present in the manuscript.
Author Response
Dear Reviewer
The authors wants to thank you for the suggestions and add the responses to you.
#7 Thank you for your clarification regarding the use of CLSI breakpoints for the VITEK AST card results. However, I would like to emphasize that there are indeed different breakpoints for antibiotics between bacteria isolated from human and animal sources. Could you please discuss how this may impact the interpretation of your resistance profiles, particularly given that your samples include both human and animal-derived isolates?
Thank you for raising this important point regarding the use of breakpoints for interpreting antibiotic resistance across isolates from different sources. In our study, the bacterial isolates were obtained from previous projects, each of which employed specific methodologies for determining resistance breakpoints tailored to the origin of the isolates.
- Human isolates: The breakpoints were determined using the clinical routines of the healthcare institutions that contributed these isolates.
- Avian isolates: Resistance profiles were established based on the methodology and criteria outlined in the referenced studies originating from the respective project.
- Environmental isolates (e.g., compost): The interpretation of resistance was based on the internal quality control protocols of the farm from which these isolates were obtained.
The highlighted paragraph was added to limitations
Our study did not establish new breakpoints but relied on these pre-existing interpretations to ensure consistency with the original analyses. The primary goal of our work was to confirm the resistance profiles reported by the contributing institutions and validate the isolates for whole-genome sequencing. Consequently, our analysis focused on identifying and characterizing the genetic elements associated with resistance as described in the manuscript.
Since the resistance interpretation was inherited from the previous studies, we are not in a position to reassess or critique the criteria employed by the original researchers. However, our findings provide a genomic perspective to complement and enhance the understanding of the resistance profiles across isolates from diverse sources.
We hope this explanation clarifies the scope and rationale of our work and its reliance on previously established methodologies.
#9 Thank you for your response regarding the VITEK AST card's capability for phenotypic ESBL confirmation. While I appreciate this point, I would like to reiterate that the gold standard for ESBL detection involves specific confirmatory tests that are essential for accurately identifying ESBL producers. Methods such as the Double Disk Synergy Test or Etest provide additional validation beyond screening methods. Additionally, the CLSI guidelines specify that ESBL testing should include Ceftazidime (0.25-128 µg/mL), Ceftazidime-clavulanate (0.25/4-128/4 µg/mL), Cefotaxime (0.25-64 µg/mL), and Cefotaxime-clavulanate (0.25/4-64/4 µg/mL). Testing necessitates using both cefotaxime and ceftazidime, both alone and in combination with clavulanate.
Thank you for raising this important point regarding the gold standard for phenotypic confirmation of ESBL production. In our study, the phenotypic confirmation, including methods like the Double Disk Synergy Test, was conducted by the original researchers who isolated the bacteria producing extended-spectrum beta-lactamases (ESBLs). These confirmations were performed under their established protocols and criteria during their initial investigations.
The highlighted paragraph was added to limitations
In our case, we utilized the VITEK AST card to reconfirm ESBL production for the specific purpose of selecting isolates that carried the genetic elements of interest for whole-genome sequencing. This approach ensured that we focused on strains containing the genes we aimed to characterize and describe in detail in this study.
The success of this method is validated by the genomic information provided in the manuscript, which aligns with the phenotypic profiles reported by the original investigators and the genetic findings of this work.
#11 I appreciate that you addressed my previous comment by separating the Limitations section. However, I noticed that the Conclusion section is still missing from the manuscript. It is important to include a distinct Conclusion section that summarizes the key findings and offers insights into clinical applications.
A conclusion section summarizing the key findings has been added to the manuscript. However, identifying new sustained clinical applications was beyond the scope of our study, as it did not specifically aim to address this aspect. Nonetheless, if the reviewer identifies relevant insights that would enhance the manuscript, we would be glad to incorporate their contributions and express our gratitude accordingly in the acknowledgment section of the paper.
The highlighted paragraph was added
Conclusions
- coli isolates of human origin belonged to ST609 and phylogroup A, while poultry and compost isolates belonged to phylogroups A, B1, E, F, and K. Diverse STs. K. pneumoniae isolates were ST13 (five isolates), ST258 (four isolates) and ST86 (one isolate). Within E. coli isolates, blaCTX-M-55, blaCTX-M-65, blaCTX-M-15, and blaCTX-M-2 genes were identified. This study also identified blaCMY-2 and blaKPC-3 (the latter in a carbapenem-susceptible isolate). In E. coli, the plasmid-borne mcr-1.1 gene was identified across all E. coli isolates within an IncI2 plasmid. Tigecycline-reduced susceptibility or resistance was related to missense-sense mutations in marA and acrA genes. Within K. pneumoiae, blaCTX-M-15 and blaCTX-M-65, blaKPC-2 and blaKPC-3, were associated to 3GC and carbapenems resistance, respectively. The blaKPC-2 allele was identified in a ̴10-kb Tn4401 transposon, with genetic context being: tnpR-tnpA-istAistB-blaKPC-2-tnpA. In K pneumoniae, sequence data and phenotypic analysis link a nonsense mutation in mgrB (K3*) and missense mutations within arnB, eptA, pmrB, pmrJ, and phoQ, to colistin resistance. While tigecycline resistance was linked to nonsense and missense mutations within the ramR gene. Additionally, this study identified several integron´s structures, including Int191(5’CS-dfrA14-3’CS), which was the most prevalent integron (Int) among E. coli and K. pneumoniae isolates, followed by Int0 (5’CS-3`CS) and Int18 (5’CS-dfrA1-3`CS). These results contribute to genomic epidemiology of MDR E. coli and K. pneumoniae at or setting and to worldwide epidemiology in a One Health approach.
#12 Regarding the updating of references, I still note a total of 55 references, and it is unclear which ones have been updated. Could you please specify which references have been revised to include more recent publications?
Although the total number of references remains unchanged at 55, the cited works have been updated and thoroughly reviewed. Each reference has been carefully evaluated not only for its publication date but also for its content. We, as the authors, believe that the current citations adequately support the manuscript. However, if the reviewer recommends the inclusion of any additional relevant work or specific references, including their own, we would be glad to incorporate them in the final version.
#13 Thank you for addressing the inconsistencies in the use of the terms "antimicrobial" and "antibiotic." However, I still noticed some discrepancies present in the manuscript.
Thank you for your valuable feedback regarding the use of the terms "antimicrobial" and "antibiotic." To avoid any potential confusion, we have replaced all instances of "antimicrobial" with "antibiotic" throughout the manuscript, as the focus of the study is specifically on antibiotics. While we acknowledge that antibiotics are a subset of antimicrobials, we believe this change clarifies the manuscript's focus on bacterial infections and antibiotic resistance. Should you have any further concerns or specific examples where the terminology may still be unclear, we would be happy to address them.
Reviewer 5 Report
Comments and Suggestions for Authors
Abstact
1. A, B1, E, F, and K. – K is redundant again
2. Diverse STs of K. pneumoniae isolates
3. “Tigecycline-reduced susceptibility or resistance was related to missense-sense mutations in marA and acrA genes.” This construction missense-sense is confusing.
4. Tn4401-this was not corrected in revised manuscript and was indicated in the first version of revision
5. tnpR-tnpA-istA-istB-blaKPC-2-tnpA-introduce the line
6. ramR
7. “at or setting” this part of the sentence was probably not corrected well.
8. Keywords are not ordered in a meaningful way. For example, they could be order like this Escherichia coli, Klebsiella pneumoniae, colistin resistance, tigecycline resistance, blaKPC, mcr-1, Ecuador
Introduction
1. 3Gx1Cs-this is probably mistake
2. the gold standard for threatening multidrug resistant (MDR) infections
3. this was not corrected-it is better to write alternation or substitutions of proteins than mutations
-tet(A) is the efflux pump
Materials and Methods
1. 4,000 Enterobacteriaceae
2. unpublished data should be in parenthesis
3. “The isolates analyzed were the total of isolates of E. coli-producing mcr-1 and carbapemen- and colistin-resistant K. pneumoniae available at our location to our study group”. This sentence is redundant since it was explained everything in previous paragraphs.
4. E. coli and K. pneumoniae should be in italics
5. “the obtained DNA was used for PCR and sequencing purposes” This should be separated sentence.
6. “Antimicrobial susceptibility testing MIC values for colistin (COL) were established by micro-dilution, using GNX2F plates (Thermo Scientific™, West Palm Beach, USA), using the minimum inhibitory concentration (MIC ≥ 2) breackpoint, and the precence of mcr-1 gene was stablished by pcr.” This sentence should be changed. For example, Antimicrobial susceptibility testing for colistin (COL) were established by microdilution method, using GNX2F plates (Thermo Scientific™, West Palm Beach, USA), while the minimum inhibitory concentration (MIC) breakpoint was 2 (here should be indicated which breakpoint table was used e.g. EUCAST). The presence of the mcr-1 gene was established by PCR method (here should be indicated reference for primers and PCR temperature condition which was used).
7. low quality – two words
8. this was not corrected- , Bogota
9. integrons instead of integros
Results
As I suggested in the previous revision this section should be separated in subsections.
1. this was not corrected -Table 1. legend (MEM), amikacin
2. In the results all E. coli were carbapenem-susceptible, but some of them harbored gene for KPC-3. How you explain this?
3. , and phoQ – and should not be italic
4. and involves the – and involved in the
5. this was not corrected - in supplementary table has been written blaCTX-M-14, so it should be checked where is the mistake
6. ramR – ramR
7. for this system – for which system?
8. int18 – Int18
9. Supplementary Material should be revised also as it was indicated previously, but I did not receive new version.
10. this was not changed - it should be indicated which gene was the most prevalent and which isolates harbored the most virulence genes (10 of them)
11. PlasmidFinder tool and conjugation experiments were not mentioned in the Materials and Methods. In previous revision, my suggestion was to explain each method by this was done only partially. So, please add this experiments as well.
12. “The mating assay and PBRT were carried out under the UNIETAR laboratory protocol.” This sentence is for Materials and Methods.
13. (Mathers et al., 2017)- the reference should be numbered
14. The figure 4 is not in accordance with the text (A, B, C, D versus A, B, C)
This part was completely ignored by the authors:
Table 2.
1. there is a mistake in the number of genes for isolate 86A (5 instead of 6); lpf vertically 5 instead of 4; iroN 2 instead of 4
2. the names of the genes should be in italics; air instead of Air, check all genes because some of the have the first letter bigger
3. the symbols for the isolates of different source in the legend are not visible
4. in the legend the names of the genes should be written in the same style like the rest of the text and explained the same for each gene e.g. iroN, enterobactin siderophore receptor protein
Figure 2A. The resolution of the figure is not of a good quality. Also, the first and the second plasmid in the figure were not mentioned in the text or legend.
Comments on the Quality of English LanguageThe English could be improved to more clearly express the research.
Author Response
We sincerely appreciate the detailed list of suggestions and corrections provided. Your thoughtful input has significantly enhanced the quality and clarity of the manuscript, and we are grateful for your valuable contribution to improving its overall coherence and impact.
Abstact
- A, B1, E, F, and K. – K is redundant again
corrected
- Diverse STs of pneumoniae isolates
corrected
- “Tigecycline-reduced susceptibility or resistance was related to missense-sense mutations in marA and acrA genes.” This construction is confusing.
corrected
- Tn4401-this was not corrected in revised manuscript and was indicated in the first version of revision
corrected
- tnpR-tnpA-istA-istB-blaKPC-2-tnpA-introduce the line
corrected
- ramR
corrected
- “at or setting” this part of the sentence was probably not corrected well.
corrected
- Keywords are not ordered in a meaningful way. For example, they could be order like this Escherichia coli, Klebsiella pneumoniae, colistin resistance, tigecycline resistance, blaKPC, mcr-1, Ecuador
corrected
Introduction
- 3Gx1Cs-this is probably mistake
corrected
- the gold standard for threatening multidrug resistant (MDR) infections
corrected
- this was not corrected-it is better to write alternation or substitutions of proteins than mutations
-tet(A) is the efflux pump
corrected
Materials and Methods
- 4,000 Enterobacteriaceae
corrected
- unpublished data should be in parenthesis
corrected
- “The isolates analyzed were the total of isolates of E. coli-producing mcr-1 and carbapemen- and colistin-resistant K. pneumoniae available at our location to our study group”. This sentence is redundant since it was explained everything in previous paragraphs.
corrected
- coli and K. pneumoniae should be in italics
corrected
- “the obtained DNA was used for PCR and sequencing purposes” This should be separated sentence.
corrected
- “Antimicrobial susceptibility testing MIC values for colistin (COL) were established by micro-dilution, using GNX2F plates (Thermo Scientific™, West Palm Beach, USA), using the minimum inhibitory concentration (MIC ≥ 2) breackpoint, and the precence of mcr-1 gene was stablished by pcr.” This sentence should be changed. For example, Antimicrobial susceptibility testing for colistin (COL) were established by microdilution method, using GNX2F plates (Thermo Scientific™, West Palm Beach, USA), while the minimum inhibitory concentration (MIC) breakpoint was 2 (here should be indicated which breakpoint table was used e.g. EUCAST). The presence of the mcr-1gene was established by PCR method (here should be indicated reference for primers and PCR temperature condition which was used).
corrected
- low quality – two words
corrected
- this was not corrected- , Bogota
corrected
- integrons instead of integros
corrected
Results
As I suggested in the previous revision this section should be separated in subsections.
added
- this was not corrected -Table 1. legend (MEM), amikacin
corrected
- In the results all E. coli were carbapenem-susceptible, but some of them harbored gene for KPC-3. How you explain this?
n was sensible to carbapenems, and the genetic environment for blaKPC-3 could not be established. Therefore, we suggest this finding as a non-functional gene.
- , and phoQ– and should not be italic
corrected
- and involves the – and involved in the
corrected
- this was not corrected - in supplementary table has been written blaCTX-M-14, so it should be checked where is the mistake
So sorry, we have not able to identify the suggestion at the supplementary file, pleas help us to correct it.
- ramR – ramR
corrected
- for this system – for which system?
corrected
- int18 – Int18
corrected
- Supplementary Material should be revised also as it was indicated previously, but I did not receive new version.
A revised version will be included for you.
- this was not changed - it should be indicated which gene was the most prevalent and which isolates harbored the most virulence genes (10 of them
added in response to your suggestion, The most prevalent virulence gene among the E. coli isolates was gad (glutamate decarboxylase). Additionally, the isolate with the highest number of virulence genes was isolate 136A, which originated from avian sources, containing the greatest diversity of these genes. However, such virulence factor profiles did not identify human related intestinal or extra-intestinal E. coli pathotypes
- PlasmidFinder tool and conjugation experiments were not mentioned in the Materials and Methods. In previous revision, my suggestion was to explain each method by this was done only partially. So, please add this experiments as well.
- “The mating assay and PBRT were carried out under the UNIETAR laboratory protocol.” This sentence is for Materials and Methods.
Mating assay and PCR-based replicon typing (PBRT)
The mating assay and PBRT were carried out under the UNIETAR laborator protocol. In brief, E coli J53 strain, resistant to sodium azide 200µg/mL was used as receptor. The mating was carried out in LB broth mixing every E. coli Isolate and the same receptor strain. Following a 18 hours at 35° C incubation. The transconjugants were selected in LB agar plates supplemented by sodium azide 200 µg/mL and colistin 2 µg/mL. The transconjugants were evaluated using colony PCR for mcr-1 gene 20 and PBRT using the PBRT 2.0 kit (Diatheva).
Plasmid Anlisys
PlasmidFinder v.2.1 coordinated with map to reference tool available in the genious prime software, and cgview_comparison_tool (https://github.com/paulstothard/cgview_comparison_tool ) were used for plasmid reconstruction.
- (Mathers et al., 2017)- the reference should be numbered
It was a mistake and was removed
- The figure 4 is not in accordance with the text (A, B, C, D versus A, B, C)
Ring alignment of K. pneumoniae ST13 isolates draft genomes against previously described plasmids pKPC-CAV1042-89 (A), and ST258 strains daft genomes against pKPC-484 (B). Alignment of Tn4401 loci from ST13 isolates against pKPC-CAV1042-89 reference (C), and ST258 strains against pKPC-484 reference (D).
This part was completely ignored by the authors:
Our apologies for the omissions
Table 2.
- there is a mistake in the number of genes for isolate 86A (5 instead of 6); lpfvertically 5 instead of 4; iroN 2 instead of 4
corrected
- the names of the genes should be in italics; air instead of Air, check all genes because some of the have the first letter bigger
corrected
- the symbols for the isolates of different source in the legend are not visible
corrected
- in the legend the names of the genes should be written in the same style like the rest of the text and explained the same for each gene e.g. iroN, enterobactin siderophore receptor protein
corrected
Figure 2A. The resolution of the figure is not of a good quality. Also, the first and the second plasmid in the figure were not mentioned in the text or legend.
Quality enhanced and legend improved
Plasmid reconstruction and alignment. Alignment of mcr-1 harboring IncI2-plasmids references pHNSHP45, pCRE10 and pSMEc189 against E. coli isolates reconstructed plasmids harboring mcr-1 genes, the zoom shows the mcr-1 neighborhood. Graphic alignment of the mcr-1.1 genetic context (A). ISApl1 transposase and mcr-1 genes are highlighted. Red arrowheads represent gaps between contigs in the draft plasmids (B).
Round 3
Reviewer 4 Report
Comments and Suggestions for Authors#7 I appreciate the clarification on how the breakpoints were tailored to the origin of the isolates. However, I would like to emphasize the importance of ensuring that the breakpoints applied are consistent and appropriate for the specific bacterial populations being studied, as discrepancies could significantly impact the interpretation of resistance profiles.
#9 Thank you for your response regarding the phenotypic confirmation of ESBL production. It would be helpful to provide a citation of the previous research that details the methodologies used for this confirmation, including the Double Disk Synergy Test, as conducted by the original researchers. This would enhance the transparency of your study and support the validity of the ESBL detection methods employed.
#11, #12 and #13 Thank you for implementing the improvements based on my previous suggestions regarding the Limitations section and the clarification on references. I appreciate your efforts, but I would still encourage you to include a distinct Conclusion section to summarize the key findings and address the remaining discrepancies in terminology throughout the manuscript.
Author Response
We sincerely appreciate your valuable suggestions and guidance during the review process. Please find attached our detailed responses to the comments and the revised manuscript, which incorporates the suggested amendments.
We remain at your disposal for any further clarifications or adjustments that may be required.
#7 I appreciate the clarification on how the breakpoints were tailored to the origin of the isolates. However, I would like to emphasize the importance of ensuring that the breakpoints applied are consistent and appropriate for the specific bacterial populations being studied, as discrepancies could significantly impact the interpretation of resistance profiles. #9 Thank you for your response regarding the phenotypic confirmation of ESBL production. It would be helpful to provide a citation of the previous research that details the methodologies used for this confirmation, including the Double Disk Synergy Test, as conducted by the original researchers. This would enhance the transparency of your study and support the validity of the ESBL detection methods employed.
Added to limitations
The strains characterized through whole-genome sequencing in this report were initially isolated in separate studies. The selection of these isolates was determined by the criteria established by the respective research groups that donated them. Consequently, this introduces inherent biases related to each study's phenotypic screening cut-offs. Given the isolates' diverse origins—spanning environmental, veterinary, and human clinical contexts—specific criteria were employed in accordance with the distinct focus of each field. Therefore, we recommend that prevalence data be interpreted with consideration of these criteria and the specific conditions of the prior studies. Nevertheless, it is important to note that for the selection of extended-spectrum β-lactamase (ESBL)-producing isolates, all studies consistently employed the double-disk synergy test, as recommended by CLSI methodology.
#11, #12 and #13 Thank you for implementing the improvements based on my previous suggestions regarding the Limitations section and the clarification on references. I appreciate your efforts, but I would still encourage you to include a distinct Conclusion section to summarize the key findings and address the remaining discrepancies in terminology throughout the
Changed conclusion
Conclusions
This study investigated nine 3GC-resistant E. coli isolates carrying the mcr-1 gene (sourced from poultry farms, human infections, and dairy-farm compost) alongside ten clinical isolates of colistin- and carbapenem-resistant K. pneumoniae. Human-origin E. coli isolates were classified as ST609 (phylogroup A), while those from poultry and compost represented phylogroups A, B1, E, and F. Diverse K. pneumoniae sequence types included ST13, ST258, and ST86. Key resistance genes identified in E. coli included blaCTX-M-55, blaCTX-M-65, blaCTX-M-15, and blaCTX-M-2, with blaCMY-2 and blaKPC-3 also detected. All mcr-1.1 genes were plasmid-borne within IncI2 plasmids. Reduced tigecycline susceptibility in E. coli was linked to marA and acrA gene mutations.
In K. pneumoniae, 3GC resistance was associated with blaCTX-M-15 and blaCTX-M-65, while carbapenem resistance was mediated by blaKPC-2 and blaKPC-3. Colistin resistance correlated with genetic alterations in mgrB, marA, acrA, arnB, eptA, pmrB, pmrJ, and phoQ, and tigecycline resistance with ramR mutations. The study also identified integron structures, with Int191 (dfrA14) being the most prevalent across isolates. These findings enhance understanding of multidrug-resistant E. coli and K. pneumoniae within the genomic epidemiology context, supporting the global One Health framework.
We sincerely thank you for your guidance throughout this process and remain available to address any further queries or concerns.
Kind regards,